# Induction of osteogenesis by bone-targeted Notch activation

**Cong Xu[1†], Van Vuong Dinh[1†], Kai Kruse[1], Hyun-Woo Jeong[1], Emma C Watson[1], Susanne Adams[1], Frank Berkenfeld[1], Martin Stehling[2], Seyed Javad Rasouli[1], Rui Fan[3], Rui Chen[3], Ivan Bedzhov[3], Qi Chen[1], Katsuhiro Kato[1], Mara E Pitulescu[1]\*, Ralf H Adams[1]\***

[1]Max Planck Institute for Molecular Biomedicine, Department of Tissue Morphogenesis, and University of Münster, Faculty of Medicine, Münster, Germany; [2]Flow Cytometry Unit, Max Planck Institute for Molecular Biomedicine, Münster, Germany; [3]Embryonic Self-Organization Research Group, Max Planck Institute for Molecular Biomedicine, Münster, Germany

**Abstract** Declining bone mass is associated with aging and osteoporosis, a disease characterized by progressive weakening of the skeleton and increased fracture incidence. Growth and lifelong homeostasis of bone rely on interactions between different cell types including vascular cells and mesenchymal stromal cells (MSCs). As these interactions involve Notch signaling, we have explored whether treatment with secreted Notch ligand proteins can enhance osteogenesis in adult mice. We show that a bone-targeting, high affinity version of the ligand Delta-like 4, termed Dll4$_{(E12)}$, induces bone formation in male mice without causing adverse effects in other organs, which are known to rely on intact Notch signaling. Due to lower bone surface and thereby reduced retention of Dll4$_{(E12)}$, the same approach failed to promote osteogenesis in female and ovariectomized mice but strongly enhanced trabecular bone formation in combination with parathyroid hormone. Single cell analysis of stromal cells indicates that Dll4$_{(E12)}$ primarily acts on MSCs and has comparably minor effects on osteoblasts, endothelial cells, or chondrocytes. We propose that activation of Notch signaling by bone-targeted fusion proteins might be therapeutically useful and can avoid detrimental effects in Notch-dependent processes in other organs.

**\*For correspondence:**
mara.pitulescu@mpi-muenster.
mpg.de (MEP);
mara.pitulescu@mpi-muenster.
mpg.de (MEP);
ralf.adams@mpi-muenster.mpg.
de (RHA)

†These authors contributed
equally to this work

**Competing interest:** The authors
declare that no competing
interests exist.

**Reviewing Editor:** Cheryl
Ackert-Bicknell, University of
Colorado, United States

## Editor's evaluation

Osteoporosis most often treated by reducing bone resorption as there are limited choices of medication that are anabolic for bone. Previous studies have suggested that the Notch signaling pathway could be targeted to enhance osteogenesis. The authors have intriguingly generated a soluble bone targeted fusion protein comprised of a modified version of the extra-cellular domain of the Delta-like 4 Notch ligand and poly-aspartate peptide motif with binding affinity for hydroxyapatite in the bone matrix. This approach has high potential as a future therapeutic for osteoporosis.

## Introduction

Osteoporosis is the most common disease affecting the skeletal system in humans and characterized by low bone mass, reduced mineral density, and disarranged bone microarchitecture. This reduces bone strength and increases the risk of fractures, which leads to secondary health problems and increased mortality (***NIH Consensus Development, 2001***; ***Cosman et al., 2014***). The skeletal system is undergoing lifelong remodeling mediated by bone-forming osteoblasts and bone-resorptive osteoclasts (***Walsh et al., 2006***). Disbalance between bone formation and resorption can lead to osteopenia,

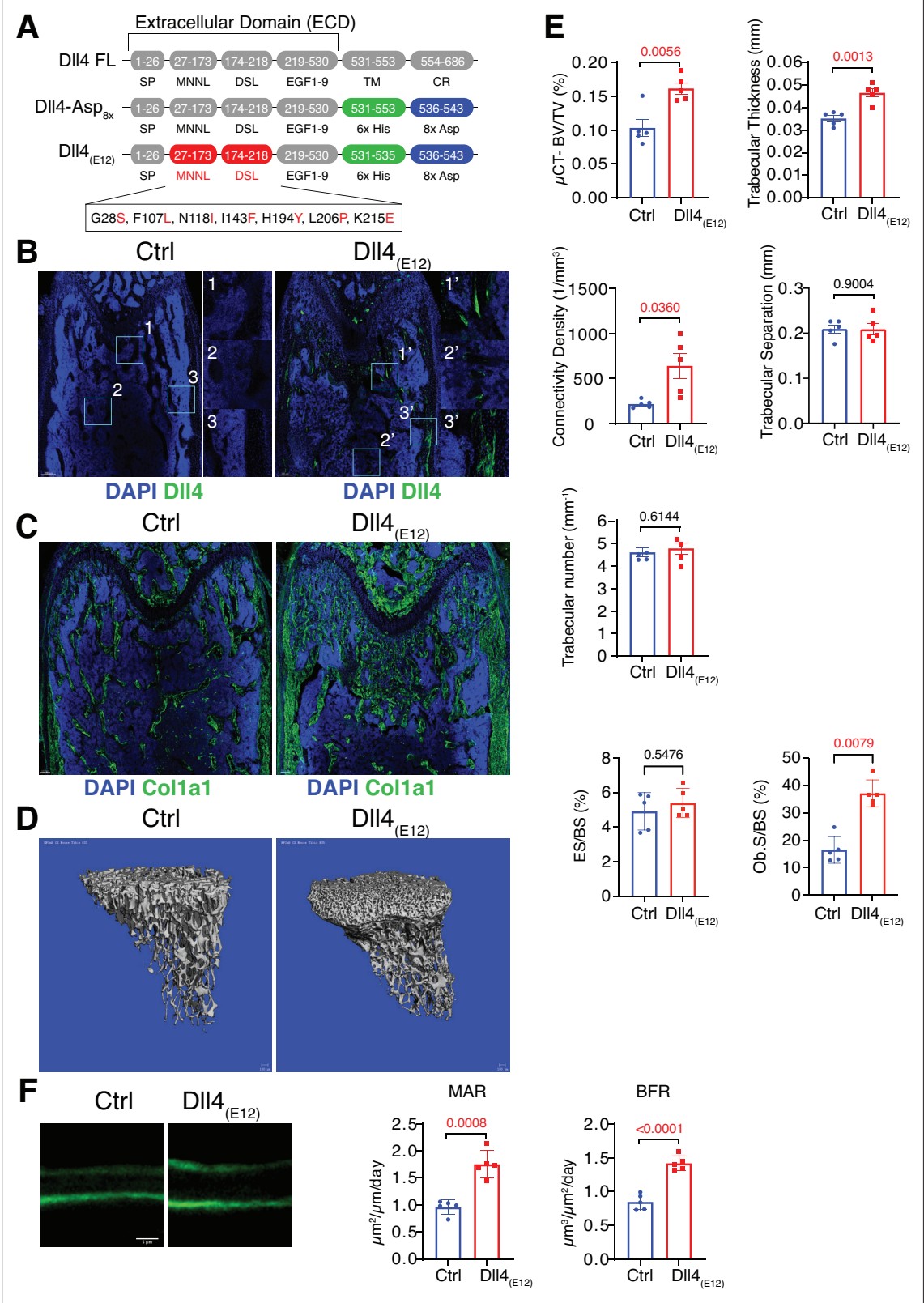

**Figure 1.** Recombinant Dll4(E12) increases bone formation in vivo. (**A**) Schematic diagram showing the domain organization of murine Dll4 full-length protein and recombinant Dll4-Asp8x and Dll4(E12) fusion proteins. The latter contain 6x His epitope tags (green boxes) and negatively charged peptides consisting of eight Asp residues (8x Asp; blue boxes) instead of the transmembrane (TM) domain and cytoplasmic region (CR) of Dll4. Missense mutations were introduced in the MNNL (modulus at the N-terminus of Notch ligands) and DSL (Delta-Serrate-Lin) domains to generate Dll4(E12) with

*Figure 1 continued on next page*

*Figure 1 continued*

increased Notch-binding affinity. The resulting amino acid replacements are highlighted in red. SP, signal peptide. (**B**) Representative overview and high-magnification confocal images of Dll4 staining (green) on the femurs of *pLIVE-Dll4*$_{(E12)}$ and control-injected mice at the age of 11 weeks. Nuclei, DAPI (blue). Images on the right show higher magnifications of insets in metaphysis (1), bone marrow (2), and cortical bone (3). (**C**) Tile scan confocal images showing Collagen I alpha one chain (Col1a1) staining (green) in sections from *pLIVE-Dll4*$_{(E12)}$ and control femur. Nuclei, DAPI (blue). (**D**) Representative 3D reconstruction of micro-computed tomography (μ-CT) measurements for tibial metaphysis of 11-week-old *pLIVE-Dll4*$_{(E12)}$- and control-injected mice. (**E**) Bone parameters measured by μ-CT analyses: bone volume/total volume (BV/TV) in percentage, trabecular thickness in millimeters, connectivity density in one per cubic millimeter, trabeculae number in one per millimeter, trabecular separation in millimeters, and trabecular number in one per millimeter. Data represent mean ± s.e.m. (n = 5 mice, except for trabecular number with n = 4 mice) (p-values determined by unpaired t-test with Welch's correction). Graphs at the bottom represent quantitation of eroded surface over bone surface (ES/BS) in percentage and osteoblast surface overs total bone surface (Ob.S/BS) in percentage, calculated from histological HE-stained bone sections. Data represent mean ± s.e.m. (n = 5 mice) (p-values determined by Mann-Whitney U test). (**F**) Representative images of calcein double labeling (7-day time interval) in mineralized sections of the distal femur confirm increased bone formation after *pLIVE-Dll4*$_{(E12)}$ injection. Quantification of mineral apposition rate (MAR) and bone formation rate (BFR) (right panels). Data represent mean ± s.e.m. (n = 5 mice) (p-values determined by unpaired t-test with Welch's correction).

The online version of this article includes the following source data and figure supplement(s) for figure 1:

**Source data 1.** Source data for *Figure 1E and F*.

**Figure supplement 1.** Generation of Dll4-Asp$_{8x}$ recombinant proteins.

**Figure supplement 1—source data 1.** Source data for *Figure 1—figure supplement 1B,C*.

**Figure supplement 1—source data 2.** Images of uncropped blots and gels with the relevant bands labeled for *Figure 1—figure supplement 1*, *Figure 1—figure supplement 4*, and *Figure 5—figure supplement 1*.

**Figure supplement 1—source data 3.** ZIP file containing the full raw unedited blots and gels for *Figure 1—figure supplement 1*, *Figure 1—figure supplement 4*, and *Figure 5—figure supplement 1*.

**Figure supplement 2.** Dll4-Asp$_{8x}$ detection in bone.

**Figure supplement 3.** Bone formation is not increased by Dll4-Asp$_{8x}$.

**Figure supplement 4.** Generation of Dll4$_{(E12)}$.

**Figure supplement 4—source data 1.** Source data for *Figure 1—figure supplement 1C,D*.

**Figure supplement 5.** Dll4$_{(E12)}$ immunoreactivity in vivo.

the loss of bone density, and frequently progresses into osteoporosis (*Feng and McDonald, 2011*). Anti-osteoporosis therapies are either anabolic, by enhancing osteoblast activity, or anti-resorptive, involving the inhibition of osteoclast activity. FDA-approved osteoclast inhibitors include bisphosphonates, such as alendronate, risedronate, ibandronate, and zoledronic acid (*Saag et al., 1998*; *Eastell et al., 2000*; *Chesnut et al., 2004*; *Lyles et al., 2007*) but also the peptide hormone calcitonin (*Chesnut et al., 2000*) and the female sex hormone estrogen (*Rossouw et al., 2002*), whereas a peptide fragment of parathyroid hormone (PTH1–34) and monoclonal antibodies inhibiting sclerostin are used in the clinic to increase bone formation (*Neer et al., 2001*; *Clarke, 2014*; *Li et al., 2009*; *Ross et al., 2014*). Apart from specific drawbacks of individual drugs, systemic administration generally facilitates the emergence of adverse side effects, which could be potentially avoided by directing therapeutic agents to bone. Examples include L-Asp-hexapeptide conjugated-estradiol (E2) (*Sekido et al., 2001*; *Yokogawa et al., 2001*) and conjugates of prostaglandin E2 and bisphosphonate (*Gil et al., 1999*). These drugs contain targeting moieties made of either synthetic bisphosphonates or repeats of negatively charged glutamate (Glu) or aspartate (Asp) amino acid residues, which show high affinity binding to hydroxyapatite, a main component of mineralized bone (*Stapleton et al., 2017*). Apart from enrichment in the skeletal system, the modified drugs show reduced plasma retention, which limits adverse effects after administration (*Sekido et al., 2001*; *Yokogawa et al., 2001*; *Katsumi et al., 2015*). However, the efficacy of these drugs still requires clinical validation and there is still high demand for novel and effective anti-osteoporosis drugs.

Notch signaling is an evolutionary conserved pathway with numerous functional roles including the regulation of osteogenesis. In mammals, there are four Notch receptors, namely Notch 1–4, and five ligands belonging either to the Jagged/Serrate (Jag1, Jag2) or the Delta-like subfamily (Dll1, Dll3, and Dll4) (*Zanotti and Canalis, 2016*). As both receptors and ligands are transmembrane proteins, their interaction requires cell-cell contact and triggers ligand internalization, which is necessary for Notch receptor activation (*Hicks et al., 2002*; *Noguera-Troise et al., 2006*; *Ramasamy et al., 2014*). Soluble fragments containing the ligand ECD can bind Notch but fail to induce receptor activation

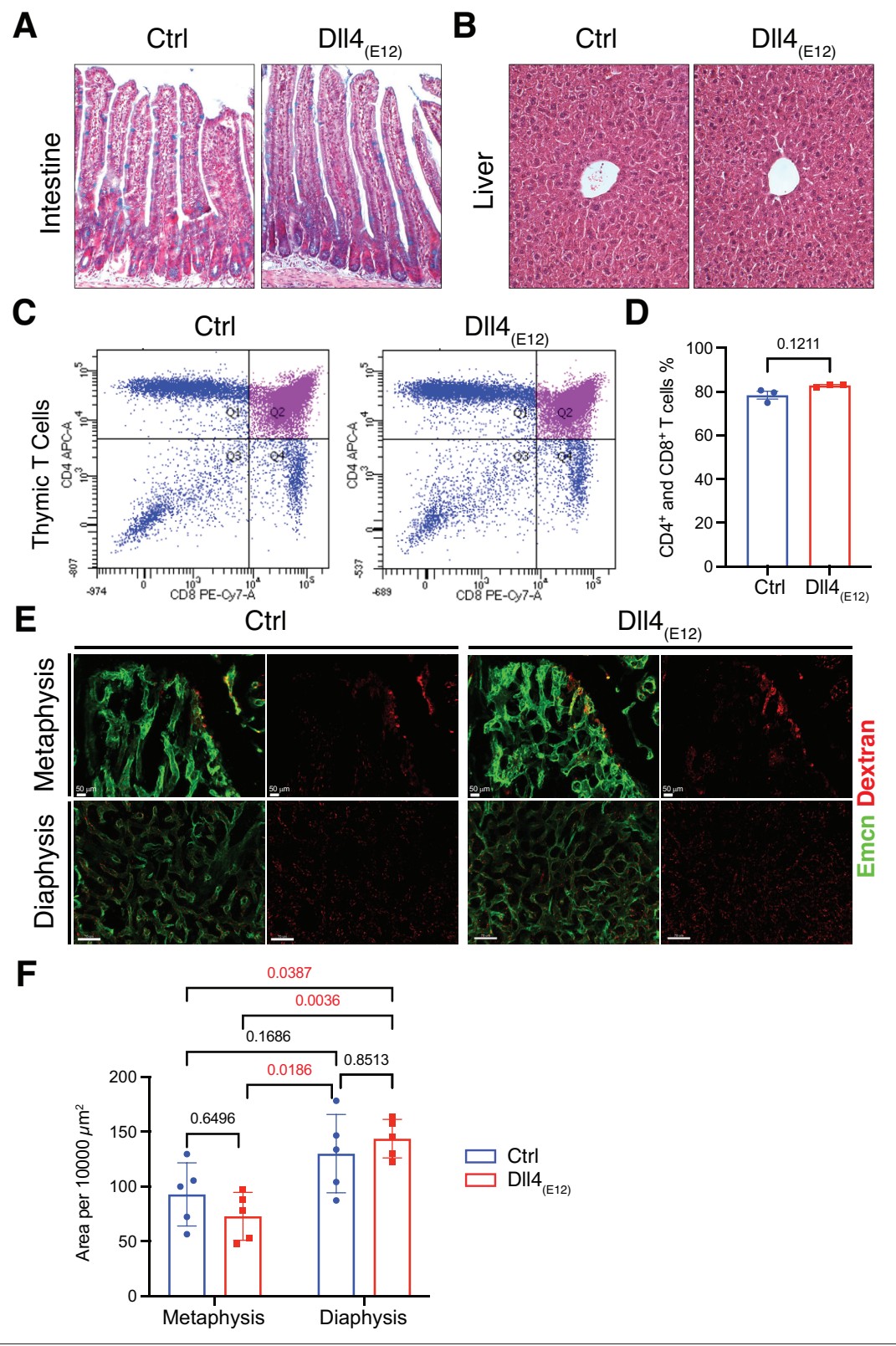

**Figure 2.** Notch-dependent processes are not blocked by Dll4(E12)in vivo. (**A**) Alcian blue and nuclear fast red double staining of small intestine. Secretory goblet cells are labeled by Alcian blue. (**B**) Hematoxylin and eosin staining of liver sections of 11-week-old *pLIVE-Dll4(E12)* and control-injected mice. (**C, D**) Analysis of CD4+/CD8+ cells from *pLIVE-Dll4(E12)* and control thymi by flow cytometry (**C**) and corresponding quantification (**D**). Data represent

*Figure 2 continued on next page*

*Figure 2 continued*

mean ± s.e.m. (n = 3 mice) (p-values determined by unpaired t-test with Welch's correction). (**E**) Confocal images showing comparable extravasation of fluorescent Dextran (70 kD) in femur of *pLIVE-Dll4*$_{(E12)}$ and control-injected mice. (**F**) Quantitation of Texas Red-labeled Dextran extravasation in femoral metaphysis and diaphysis of *pLIVE-Dll4*$_{(E12)}$ and control-injected mice. Data represent mean ± s.e.m. (n = 5 mice) (adjusted p-values determined by two-way ANOVA with Tukey's multiple comparison test).

The online version of this article includes the following source data and figure supplement(s) for figure 2:

**Source data 1.** Source data for *Figure 2D and F*.

**Figure supplement 1.** Inflammatory cytokines are not induced by Dll4$_{(E12)}$.

**Figure supplement 1—source data 1.** Source data for *Figure 2—figure supplement 1*.

**Figure supplement 2.** Dll4$_{(E12)}$ does not induce vascular leakage.

**Figure supplement 2—source data 1.** Source data for *Figure 2—figure supplement 2A-C*.

and thereby effectively act as antagonists (*Hicks et al., 2002*; *Noguera-Troise et al., 2006*). In skeletal development, Notch signaling in limb mesenchyme controls the maintenance and proliferation of mesenchymal progenitor cells and prevents premature osteoblastic differentiation (*Hilton et al., 2008*; *Engin et al., 2008*). Context-specific Notch signaling also plays important roles in osteoclast differentiation and function (*Yu and Canalis, 2020*). Notch activity in bone marrow (BM) macrophages inhibits commitment to osteoclast differentiation, but it enhances the maturation and function of committed osteoclast precursors (*Jiménez-Alcázar et al., 2017*; *Wu et al., 2010*). Notch is generally a negative regulator of endothelial cell (EC) proliferation and vascular growth, but exerts the opposite function in bone and promotes the formation of type H vessels. This capillary subtype is associated with osteoblast lineage cells and enhances osteogenesis by providing growth factors and other signals (*Ramasamy et al., 2014*; *Kusumbe et al., 2014*; *Polacheck et al., 2017*; *Langen et al., 2017*).

The reports above suggest that Notch activation, for example, through the administration of exogenous ligand molecules, might have the capacity to enhance bone formation. Given the many functional roles of Notch signaling in different organs and processes, it would be desirable to preferentially direct soluble Notch ligands to bone and thereby limit potentially harmful side effects elsewhere in the body. Here, we report the generation of a soluble, bone-targeted fusion protein, termed Dll4$_{(E12)}$, combining the extracellular domain (ECD) of Delta-like 4 (Dll4), several point mutations to increase affinity for Notch receptors (*Luca et al., 2015*), and a negatively charged poly-aspartate peptide motif (Asp$_{8x}$) mediating binding to hydroxyapatite. We show that Dll4$_{(E12)}$ increases bone formation in male adult mice without inducing adverse effects in other organs that have been previously linked to compromised Notch function. The same approach fails to enhance osteogenesis in female and ovariectomized mice, which we attribute to the lower bone surface and reduced retention of bone-targeted ligand in these animals. In contrast, bone formation in response to parathyroid hormone (PTH) is strongly enhanced by Dll4$_{(E12)}$ in female mice. Based on the histological analysis of Dll4$_{(E12)}$-treated tissue samples as well as immunohistological and single cell RNA sequencing (scRNA-seq) analysis of bone stromal cell populations, we conclude that Dll4$_{(E12)}$ is safe and can be used to activate Notch signaling in bone mesenchymal stromal cells (MSCs) leading to enhanced osteogenesis.

## Results

### Generation and characterization of soluble, bone-binding Dll4

Notch signaling plays important roles in the maintenance of mesenchymal progenitor cells and the coupling of angiogenesis and osteogenesis in the developing skeletal system (*Ramasamy et al., 2014*; *Hilton et al., 2008*; *Engin et al., 2008*; *Kusumbe et al., 2014*). To explore whether Notch activation might be therapeutically useful to enhance bone formation in the adult organism, we generated a fusion construct comprising the ECD of Dll4, a critical Notch ligand in ECs, a poly-histidine epitope tag (His$_{6x}$) enabling protein purification and detection, and a carboxyterminal stretch of eight aspartic acid residues (Asp$_{8x}$) mediating binding to hydroxyapatite (Dll4-Asp$_{8x}$) (*Figure 1A*). Expression and molecular weight of the secreted protein was validated by expression in HEK293 cells followed by sodium dodecyl sulfate polyacrylamide gel electrophoresis (SDS-PAGE) of the culture supernatant (*Figure 1—figure supplement 1A*). Previous work has established that soluble Notch ligands

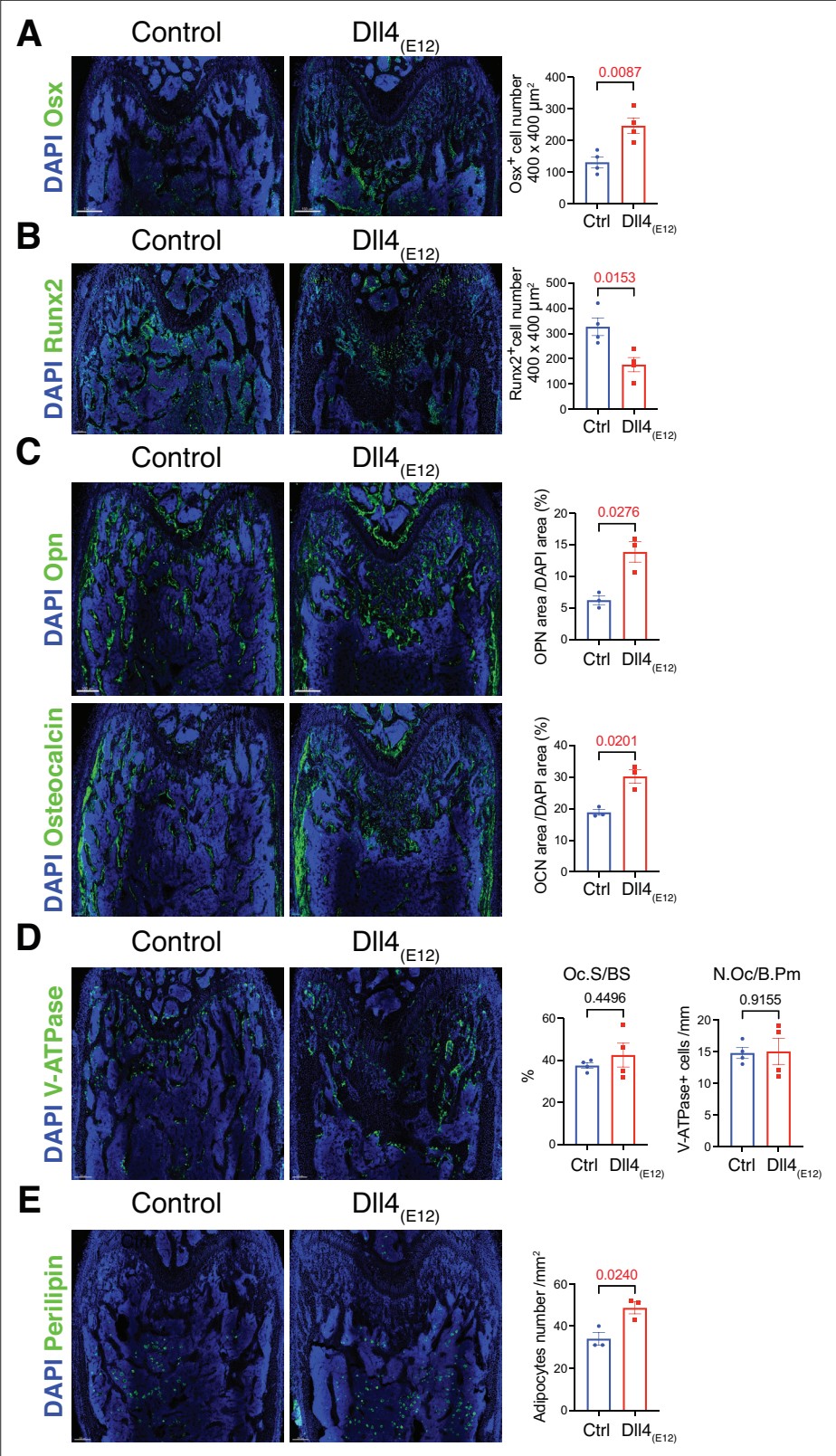

**Figure 3.** Recombinant Dll4(E12) increases osteogenesis. (**A**) Tile scan confocal images of Osterix (OSX) staining (green) in femurs from *pLIVE-Dll4(E12)* and control-injected mice. Nuclei, DAPI (blue). Graph shows quantitation of OSX+ cells. Data represent mean ± s.e.m. (n = 4 mice) (p-values determined by two-tailed unpaired t-test). (**B**) Runx2 staining (green) of *pLIVE-Dll4(E12)* and control femurs. Nuclei, DAPI (blue). Graph shows quantitation

*Figure 3 continued on next page*

*Figure 3 continued*

of Runx2$^+$ cells. Data represent mean ± s.e.m. (n = 4 mice) (p-values determined by two-tailed unpaired t-test). (**C**) Representative images showing Osteopontin (Opn) and Osteocalcin staining (green) in *pLIVE-Dll4$_{(E12)}$* and control femur. Nuclei, DAPI (blue). Data represent mean ± s.e.m. (n = 3 mice) (p-values determined by unpaired t-test with Welch's correction). (**D**) Tile scan images of ATP6V1B1+ ATP6V1B2 (V-ATPase) staining (green) in *pLIVE-Dll4$_{(E12)}$* and control femur. Nuclei, DAPI (blue). Graphs show quantification of osteoclast surface/bone surface (Os. S/B S) and osteoclast number/bone perimeter (No. Oc./B. Pm). Data represent mean ± s.e.m. (n = 4 mice) (p-values determined by unpaired t-test with Welch's correction). (**E**) Tile scan images of Perilipin staining (green) in *pLIVE-Dll4$_{(E12)}$* and control femurs. Nuclei, DAPI (blue). Graph shows quantitation of Perilipin$^+$ adipocytes. Data represent mean ± s.e.m. (n = 3 mice) (p-values determined by unpaired t-test with Welch's correction).

The online version of this article includes the following source data and figure supplement(s) for figure 3:

**Source data 1.** Source data for *Figure 3A–E*.

**Figure supplement 1.** Activation of the *Hey1-EGFP* reporter by Dll4$_{(E12)}$.

**Figure supplement 2.** Effect of Dll4$_{(E12)}$ on metaphyseal mesenchymal stromal cells (MSCs) and chondrocytes.

**Figure supplement 2—source data 1.** Source data for *Figure 3—figure supplement 2A-D*.

lack the capacity to activate their corresponding receptors and act as inhibitors blocking the interaction between endogenous, membrane-anchored Delta/Jagged ligands and Notch receptors (***Hicks et al., 2002***; ***Noguera-Troise et al., 2006***). As expected, Dll4-Asp$_{8x}$ inhibits the expression of established Notch target genes in confluent human umbilical vascular endothelial cells (HUVECs) in vitro (***Figure 1—figure supplement 1B***). When culture dishes were pre-coated with poly-L-lysine enabling the binding of negatively charged molecules, Dll4-Asp$_{8x}$ acts as an activator and induces the transcription of Notch target genes in sub-confluent HUVECs (***Figure 1—figure supplement 1C***). These data indicate that Dll4-Asp$_{8x}$ can interact with Notch receptors and that immobilization via the Asp$_{8x}$ motif enables Notch activation in cultured cells.

For in vivo experiments in mice, a cDNA encoding Dll4-Asp$_{8x}$ was cloned into the vector *pLIVE*, which mediates constitutive protein expression in liver after hydrodynamic tail vein injection (***Jiménez-Alcázar et al., 2017***). Immunohistological characterization of sectioned *pLIVE-Dll4-Asp$_{8x}$*-treated femurs isolated from male mice at 2 days post-injection (dpi) and 3 weeks post-injection (wpi) shows strong Dll4 signals at trabecular bone, the endosteum lining the inner surface of compact bone, and around distal vessel buds in proximity of the growth plate. In contrast, no comparable signals are seen in samples from male control animals (***Figure 1—figure supplement 2A-C*** and ***Figure 1—figure supplement 3A***). These data confirm the successful expression of recombinant Dll4-Asp$_{8x}$ in vivo and binding of the fusion protein to bone. However, Dll4-Asp$_{8x}$-treated bone samples at 3 wpi do not show overt alterations in Osterix$^+$ osteoblast lineage cells and deposition of the matrix proteins Osteopontin (Opn) and Collagen I alpha one chain (Col1a1) (***Figure 1—figure supplement 3B***).

## Generation of a high affinity Dll4 fusion protein

To address the possibility that Dll4-Asp$_{8x}$ does not achieve sufficient levels of Notch activation in bone, we also generated a high affinity variant of this protein by introducing multiple point mutations into the Dll4 ECD, which were previously reported to enhance Notch binding and signaling (***Luca et al., 2015***; ***Figure 1A*** and ***Figure 1—figure supplement 4A***). Following expression in HEK293 cells, SDS-PAGE confirmed the correct molecular weight of the resulting Dll4$_{(E12)}$ version of Dll4-Asp$_{8x}$ in culture supernatants (***Figure 1—figure supplement 4B***). Purified and soluble Dll4$_{(E12)}$ inhibits endogenous Notch signaling in cultured HUVECs similar to Dll4-Asp$_{8x}$, whereas Dll4$_{(E12)}$ in combination with poly-L-lysine pre-coating induces the upregulation of Notch target genes (***Figure 1—figure supplement 4C and D***). To overexpress Dll4$_{(E12)}$ in vivo, a cDNA encoding Dll4$_{(E12)}$ was cloned into the *pLIVE* vector and the resulting *pLIVE-Dll4$_{(E12)}$* construct was administered to male mice via hydrodynamic tail vein injection. Western blot analysis of liver lysates confirmed the expression of Dll4$_{(E12)}$ at 3 wpi (***Figure 1—figure supplement 4E***). Consistent with the hydroxyapatite-binding properties of Dll4-Asp$_{8x}$, immunohistological analysis of femurs confirms the accumulation of Dll4$_{(E12)}$ at the surface of trabecular and cortical bone (***Figure 1B***). Notably, histological analysis also reveals a strong increase in the Col1a1-positive area in *pLIVE-Dll4$_{(E12)}$*-injected mice (***Figure 1C***). Micro-computed tomography (μ-CT) and histomorphometric analysis show significant increases in bone mass and density (bone volume over total volume [BV/TV]), connectivity density and thickness of trabeculae and osteoblast

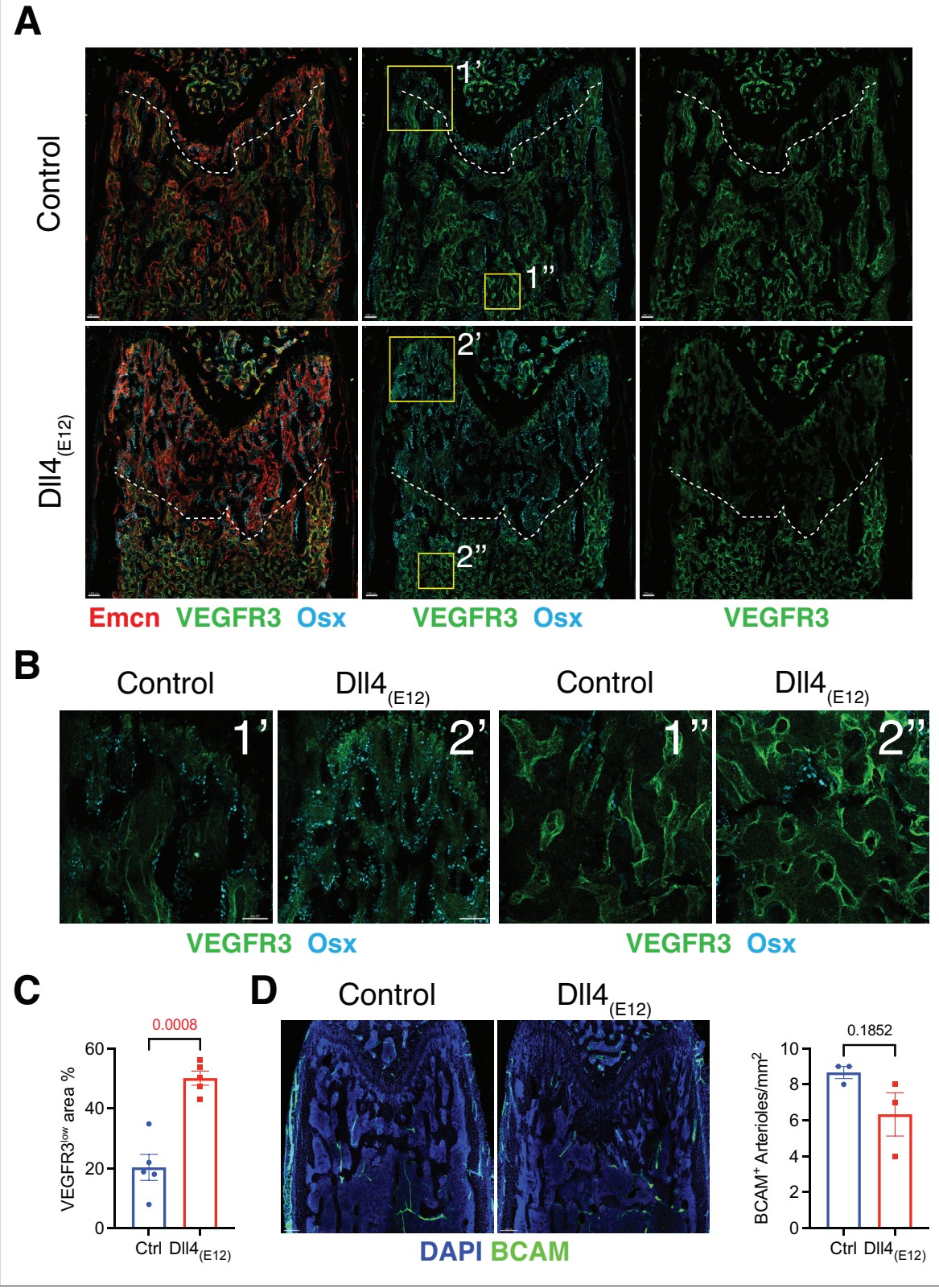

**Figure 4.** Effect of Dll4(E12) on bone vasculature. (**A**) Tile scan images of Emcn (red), VEGFR3 (green), and Osterix (OSX)- (cyan) stained sections of *pLIVE-Dll4(E12)* and control femur at 11 weeks of age. Dashed line indicates the border of VEGFR3high and VEGFR3low area. (**B**) Higher magnifications of insets in (**A**) showing the metaphysis close to growth plate (1' and 2') and bone marrow close to transition zone (1" and 2"). Stainings show VEGFR3 (green) and OSX (cyan). Note increase in OSX+ cells and presence of OSX-stained nuclei close to VEGFR3high vessels after Dll4(E12) treatment. (**C**) Quantification

*Figure 4 continued on next page*

_Figure 4 continued_

of VEGFR3$^{low}$ area per region of interest (ROI) in the metaphysis. Data represent mean ± s.e.m. (n = 5 mice) (p-values determined by unpaired t-test with Welch's correction). (**D**) Tile scan images of BCAM (green) stained sections of _pLIVE-Dll4$_{(E12)}$_ and control femur. Graph shows quantification of the number of BCAM$^+$ arteries per ROI in the metaphysis. Data represent mean ± s.e.m. (n = 3 mice) (p-values determined by unpaired t-test with Welch's correction).

The online version of this article includes the following source data for figure 4:

**Source data 1.** Source data for _Figure 4C and D_.

surface in _pLIVE-Dll4$_{(E12)}$_-injected tibiae (_**Figure 1D and E**_). However, no significant changes are observed in trabecular separation, trabecular number, and erosion surface (_**Figure 1E**_). In vivo double calcein labeling further argues for enhanced bone formation in mice treated with Dll4$_{(E12)}$ (_**Figure 1F**_). Together, these results show that key parameters of bone quality can be enhanced by expression of a soluble, hydroxyapatite-binding version of Dll4 with high affinity for Notch receptors.

## Lack of adverse biological effects of Dll4$_{(E12)}$ treatment

The Notch pathway is involved in a wide range of biological processes and, accordingly, Notch inhibition in vivo can have serious pathological effects. Considering that circulating Dll4$_{(E12)}$ released by liver cells might act as an antagonist of endogenous Notch signaling interactions, we examined several different organs for Dll4 immunoreactivity and the appearance of pathological alterations. While liver hepatocytes show the expected expression of Dll4$_{(E12)}$ (_**Figure 1—figure supplement 5A**_), no increase in Dll4 immunoreactivity indicating binding of Dll4$_{(E12)}$ was seen in spleen or lung (_**Figure 1—figure supplement 5B and C**_). In small intestine, Notch inhibition has been linked to toxicity by impairing the formation and distribution of goblet cells (_**Wu et al., 2010**_). However, intestinal crypts from _pLIVE-Dll4$_{(E12)}$_-injected mice show no goblet cell metaplasia (_**Figure 2A**_). While chronic Dll4 blockade was shown to cause sinusoidal vessel dilation in liver (_**Yan et al., 2010**_), histological analysis reveals no differences between _pLIVE-Dll4$_{(E12)}$_ and control livers (_**Figure 2B**_). This result is particularly remarkable, as liver is the source of Dll4$_{(E12)}$ protein expression after hydrodynamic tail vein injection. Dll4-mediated Notch1 activation is essential for thymic T cell development (_**Wu et al., 2010**_), but flow cytometry shows comparable numbers of CD4$^+$ and CD8$^+$ double positive T cells in thymi from control and _pLIVE-Dll4$_{(E12)}$_-injected mice (_**Figure 2C and D**_). While it has been proposed that Dll4-triggered Notch signaling modulates inflammatory responses (_**Fung et al., 2007**_), inflammatory cytokines are also not significantly increased in plasma from _pLIVE-Dll4$_{(E12)}$_-injected mice (_**Figure 2—figure supplement 1**_). Notch inhibition was also shown to compromise the barrier function of vascular endothelium (_**Polacheck et al., 2017**_). However, the extravasation of fluorescent Texas-Red-labeled Dextran is not increased in femurs from _pLIVE-Dll4$_{(E12)}$_-injected mice (_**Figure 2E and F**_). Furthermore, there is no significant difference in Texas Red Dextran extravasation in other organs investigated, namely liver, spleen, and lung (_**Figure 2—figure supplement 2A-C**_). The sum of these results indicates that Dll4$_{(E12)}$ causes no adverse systemic effects that are known to result from Notch inhibition.

## Effects of recombinant Dll4$_{(E12)}$ on bone and vasculature

Next, we performed a more detailed analysis of the alterations in _pLIVE-Dll4$_{(E12)}$_ femur at 3 wpi. While Osterix$^+$ (OSX$^+$) osteoblasts are strikingly increased relative to control samples, there is a significant reduction in the number of Runx2$^+$ osteoprogenitors (_**Figure 3A and B**_ and _**Figure 3—figure supplement 1**_). Dll4$_{(E12)}$ treatment was also performed on _Tg(Hey1-EGFP)$^{ID40Gsat}$_ mice, which express enhanced green fluorescent protein (GFP) under the control of _Hey1_, a Notch target gene. Dll4$_{(E12)}$ strongly increases the number of perivascular GFP$^+$ cells in the metaphysis relative to control animals (_**Figure 3—figure supplement 1**_). Notably, these cells are located in close proximity to OSX$^+$ osteoblasts but lack strong nuclear OSX immunostaining. Vessel-associated Hey1-expressing cells in Dll4$_{(E12)}$-treated mice are also found in the transition zone interconnecting the metaphyseal and diaphyseal vasculature, whereas no GFP$^+$ cells are found in BM (_**Figure 3—figure supplement 1**_). In addition to the increase in OSX$^+$ cells, Dll4$_{(E12)}$ treatment enhances the expression of major non-collagenous proteins involved in bone matrix organization and deposition, namely Opn and Osteocalcin (_**Figure 3C**_). Similarly, immunostaining of the receptor tyrosine kinase PDGFRβ and the proteoglycan NG2, both markers of MSCs, are significantly increased in the _pLIVE-Dll4$_{(E12)}$_ metaphysis (_**Figure 3—figure supplement 2A, B**_). Arguing against increased bone turnover involving higher bone resorption, Dll4$_{(E12)}$ treatment does

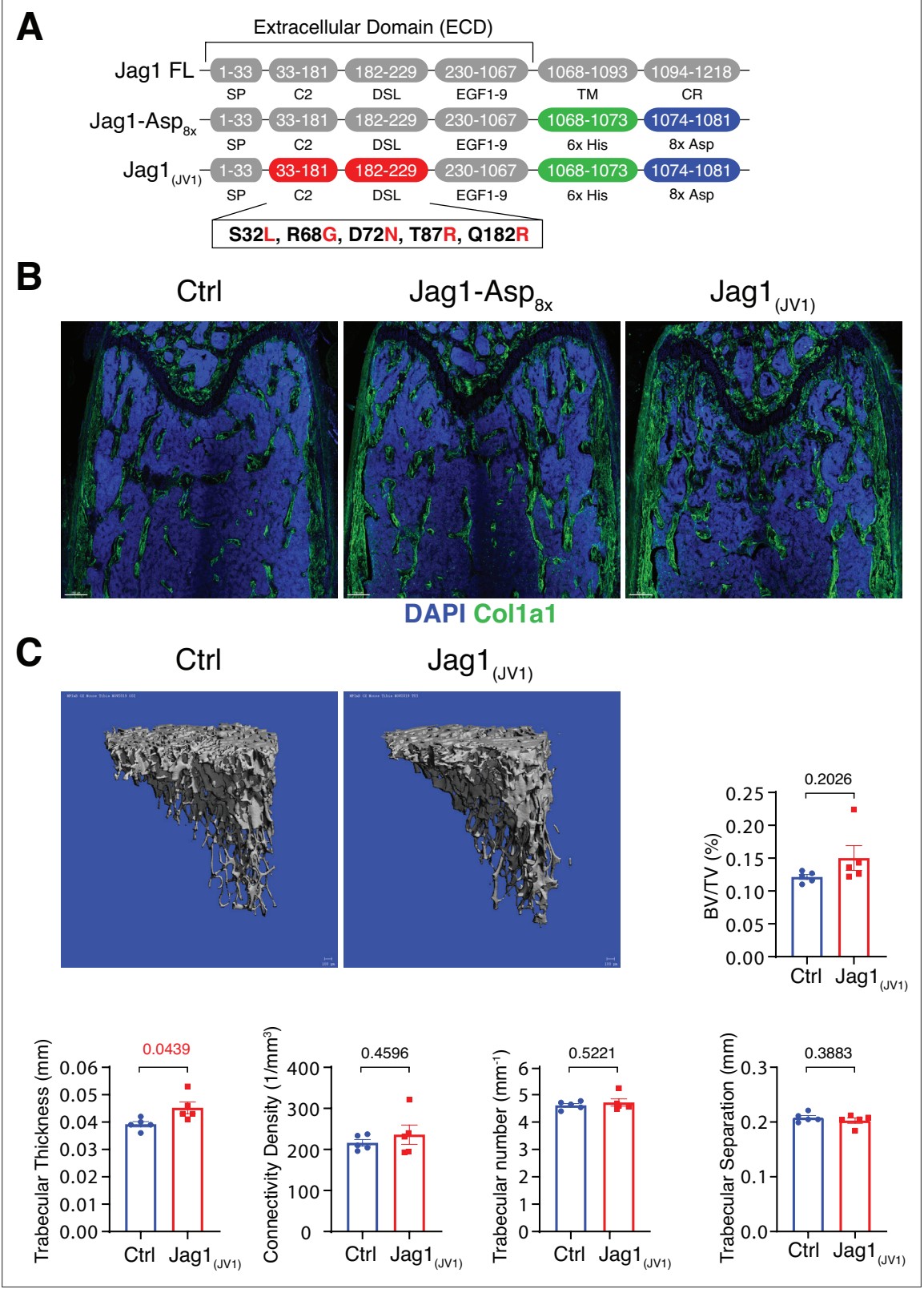

**Figure 5.** Bone formation is not changed by recombinant Jag1-Asp$_{8x}$ and Jag1$_{(JV1)}$. (**A**) Schematic diagram showing the domain organization of murine Jag1 full-length protein and recombinant Jag1-Asp$_{8x}$ and Jag1$_{(JV1)}$ fusion proteins. Fusion proteins contain 6x His epitope tag (green boxes) and Asp$_{8x}$ negatively charged peptide motif (blue boxes) instead of the transmembrane (TM) domain and cytoplasmic region (CR) of Jag1. Missense mutations were introduced in the calcium-binding C2 and DSL (Delta-Serrate-Lin) domains to generate Jag1$_{(JV1)}$ with increased Notch binding affinity. The resulting

*Figure 5 continued on next page*

*Figure 5 continued*

amino acid replacements are highlighted in red. SP, signal peptide. (**B**) Tile scan images of Collagen I alpha one chain (Col1a1) (green) staining on the femur sections of control, *pLIVE-Jag1-Asp$_{8x}$* and *pLIVE-Jag1$_{(JV1)}$*-injected mice at the age of 11 weeks. Nuclei, DAPI (blue). (**C**) Representative 3D reconstruction from micro-computed tomography (μ-CT) measurements of tibial metaphysis of *pLIVE-Jag1$_{(JV1)}$* and control-injected mice. Diagrams show bone parameters measured in μ-CT analyses: bone volume/total volume (BV/TV) in percentage, trabecular thickness in millimeters, connectivity density in one per cubic millimeter, trabeculae number in one per millimeter, and trabecular separation in millimeters. Data represent mean ± s.e.m. (n = 5 mice), (p-values determined by unpaired t-test with Welch's correction).

The online version of this article includes the following source data and figure supplement(s) for figure 5:

**Source data 1.** Source data for *Figure 5C*.

**Figure supplement 1.** Generation of Jag1-Asp$_{8x}$ and Jag1$_{(JV1)}$ recombinant proteins.

**Figure supplement 1—source data 1.** Source data for *Figure 5—figure supplement 1D,E*.

**Figure supplement 2.** Detection of Jag1-Asp$_{8x}$ and Jag1$_{(JV1)}$ in the bone.

not lead to significant alterations in the number of V-ATPase⁺ osteoclasts (*Figure 3D*). Furthermore, immunohistological analysis reveals that adipocyte number and the expression of critical chondrocyte markers, such as Aggrecan (Acan) and the transcription factor Sox9, are not significantly altered by recombinant Dll4$_{(E12)}$ (*Figure 3E* and *Figure 3—figure supplement 2C, D*).

To assess whether blood vessels in long bone are altered by treatment with Dll4$_{(E12)}$, we examined the expression of some known vascular makers such as Endomucin (Emcn) and VEGFR3 in femoral sections. Low VEGFR3 immunostaining was previously shown to mark type H vessels columns that are associated with OSX⁺ osteoprogenitors in the metaphysis, whereas vessel buds in direct proximity of growth plate chondrocytes and the sinusoidal (type L) vasculature of the BM show high anti-VEGFR3 signal (*Langen et al., 2017*). Notably, the region with VEGFR3$^{low}$ (type H) Emcn⁺ vasculature and OSX⁺ cells is significantly increased in the metaphysis of *pLIVE-Dll4$_{(E12)}$*-injected mice relative to control (*Figure 4A–C*). Notch signaling in bone ECs was previously shown to lead to an expansion of type H vasculature and increased osteogenesis, but it also promotes artery formation (*Ramasamy et al., 2014*). However, the abundance of arteries and small arterioles is not significantly altered by Dll4$_{(E12)}$ (*Figure 4D*).

## Bone formation is not increased by Jag1-Asp$_{8x}$ and Jag1$_{(JV1)}$

Homotypic signaling interactions between Notch receptors and the ligand Jagged1 in osteochondral progenitors negatively regulate the pool of these progenitors but also promote bone formation (*Youngstrom et al., 2016*; *Liu et al., 2016*). Moreover, Delta and Jagged ligands have distinct, sometimes opposite biological functions in certain processes such as retinal angiogenesis or pancreas development (*Benedito et al., 2009*; *Golson et al., 2009*). To address whether exogenous Jagged1 has the capacity to enhance bone formation, we generated a soluble Jag1-Asp$_{8x}$ fusion protein in which, analogous to Dll4-Asp$_{8x}$, the transmembrane and intracellular domain are replaced by His$_{6x}$ and Asp$_{8x}$ sequence motifs. In parallel, a high affinity variant of Jag1-Asp$_{8x}$, termed Jag1$_{(JV1)}$, was generated by replacing several amino acid residues in the ligand ECD, as previously reported (*Luca et al., 2017*; *Figure 5A*). Jag1-Asp$_{8x}$ and Jag1$_{(JV1)}$ proteins were purified from HEK293 cell culture supernatants. SDS-PAGE and Western blotting confirmed the correct molecular weight of the purified proteins (*Figure 5—figure supplement 1A-C*). Soluble Jag1$_{(JV1)}$ inhibits *DLL4, HEY1, HEY2*, and *EFNB2*, known targets of Notch signaling, in cultured HUVECs, whereas immobilized Jag1$_{(JV1)}$ induces upregulation of Notch target genes at higher levels compared to Jag1-Asp$_{8x}$ stimulation (*Figure 5—figure supplement 1D and E*).

To overexpress these recombinant proteins in vivo, cDNAs of *Jag1-Asp$_{8x}$* and *Jag1$_{(JV1)}$*, respectively, were cloned into *pLIVE* vector. Following hydrodynamic tail vein injection of male mice, both Jag1-Asp$_{8x}$ and Jag1$_{(JV1)}$ are readily detectable at bone surfaces in sections, but Jag1-Asp$_{8x}$ is also more broadly detected throughout the marrow cavity (*Figure 5—figure supplement 2A,B*). Neither Jag1-Asp$_{8x}$ nor Jag1$_{(JV1)}$, however, induce appreciable alterations in Col1a1 deposition (*Figure 5B*). μ-CT analysis reveals a slight but significant increase in trabecular bone thickness in *pLIVE-Jag1$_{(JV1)}$*-injected mice, whereas BV/TV, connectivity density, trabecular number, and trabecular separation are not altered significantly (*Figure 5C*). These results argue that only Dll4$_{(E12)}$ is able to stimulate substantial osteogenesis in adult mice, while Dll4-Asp$_{8x}$, Jag1-Asp$_{8x}$, and Jag1$_{(JV1)}$ lack this capacity.

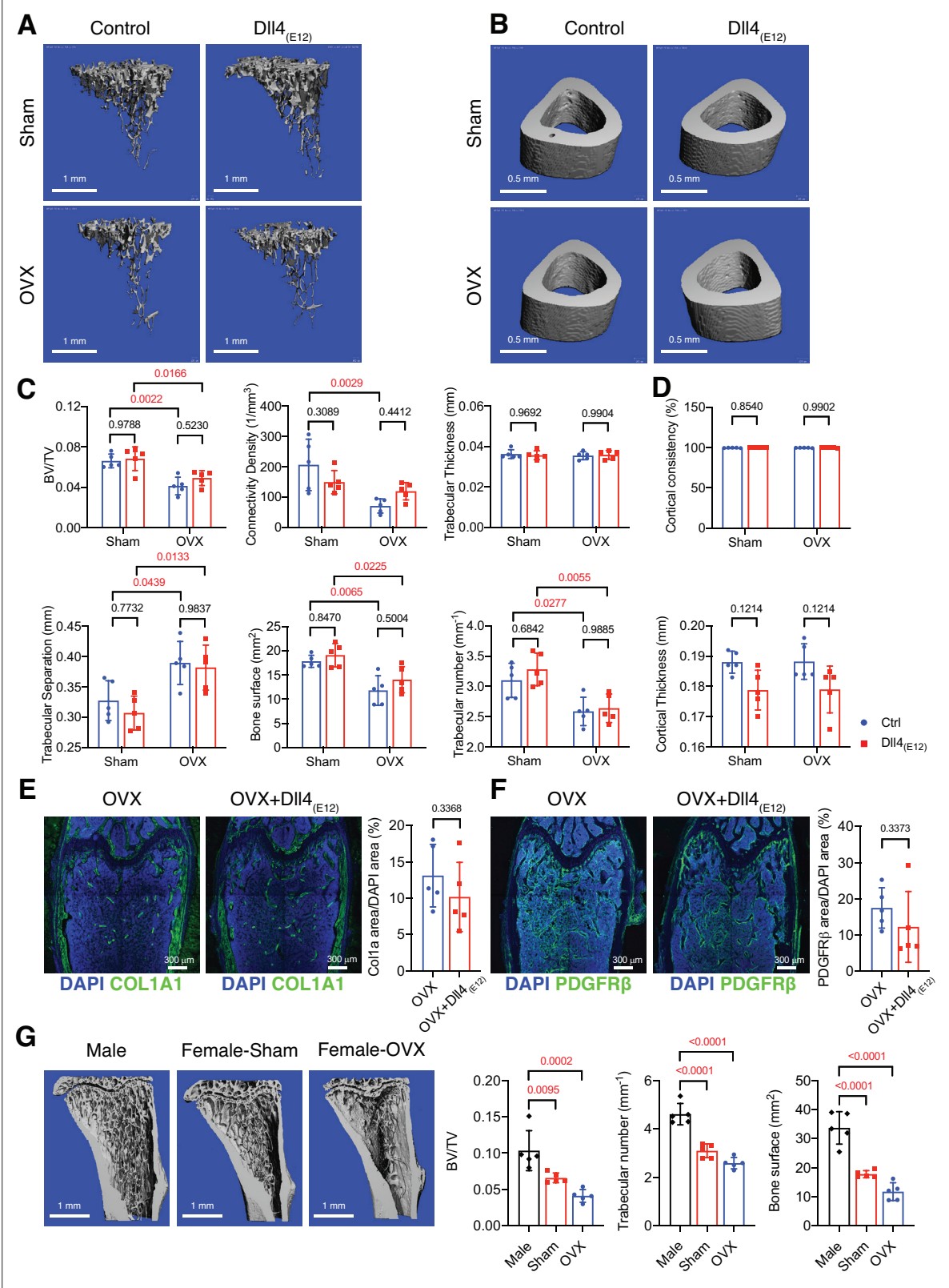

**Figure 6.** Bone formation is not increased by Dll4(E12) in ovariectomized female mice. (**A, B**) Representative 3D micro-computed tomography (μ-CT) images of trabecular bone in the distal tibial metaphysis (**A**) and cortical bone in the mid-tibial diaphysis (**B**) of sham or ovariectomized mice treated with vehicle or *pLIVE-Dll4(E12)*. (**C**) Bone parameters measured by μ-CT analysis of trabecular bone volume/total volume, trabecular connectivity density, trabecular thickness, trabecular separation, bone surface, and trabecular number in the distal tibial metaphysis. Data represent mean ± SD. (n = 5 mice)

*Figure 6 continued on next page*

*Figure 6 continued*

(p-values determined by two-way ANOVA with Tukey's multiple comparisons test). (**D**) Quantitation of μ-CT analysis of cortical bone consistency and cortical thickness in the mid-tibial diaphysis. Data represent mean ± SD. (n = 5 mice) (p-values determined by two-way ANOVA with Tukey's multiple comparisons test). (**E, F**) Tile scan confocal images showing Collagen I alpha one chain (Col1a1) staining (green; **E**) and PDGFRβ staining (green; **F**) in femoral sections from ovariectomized female mice treated with vehicle or *pLIVE-Dll4(E12)*. Nuclei, DAPI (blue). Graphs show relative ratio (percentage) of Col1a area/DAPI area and PDGFRβ area/DAPI area. Data represent mean ± SD. (n = 5 mice) (p-values determined by two-tailed unpaired t-test (**E**) with Welch's correction (**F**)). (**G**) Representative μ-CT images of trabecular bone in the distal tibial metaphysis of male, female sham, and ovariectomized mice. Graphs show quantitation of the trabecular bone volume/total volume, trabecular number, and bone surface. Data represent mean ± SD. (n = 5 mice), (p-values determined by one-way ANOVA followed by Sidak's multiple comparisons test).

The online version of this article includes the following source data and figure supplement(s) for figure 6:

**Source data 1.** Source data for *Figure 6C–G*.

**Figure supplement 1.** Reduced Dll4(E12) immunoreactivity and trabecular bone in female mice compared to male mice.

**Figure supplement 1—source data 1.** Source data for *Figure 6—figure supplement 1A-D*.

## Sex-specific effects of Dll4(E12) on bone formation

Next, we investigated whether Dll4(E12) treatment provides any beneficial effects in the ovariectomy model of postmenopausal osteoporosis. In comparison to sham controls, ovariectomized female mice show significant reduction of trabecular bone, which is not improved by Dll4(E12) expression (*Figure 6A–D*). Similarly, Col1a1 and PDGFRβ immunoreactivity is not significantly altered by Dll4(E12) (*Figure 6E–F*).

Given that bone surface is substantially lower in adult female mice relative to age-matched males (*Figure 6G* and *Figure 6—figure supplement 1A,B*), we reasoned that Dll4(E12) binding might depend on the sex of animals. Indeed, endogenous Dll4 immunoreactivity is already significantly lower in control females relative to males and the strong increase in femur sections from Dll4(E12)-expressing males is not mirrored by female samples (*Figure 6—figure supplement 1C, D*). Together, these findings indicate that the biological effect of Dll4(E12) depends on the available bone surface, suggesting that this approach might not be beneficial in settings with low bone mass.

Anabolic therapy is a clinical approach for the treatment of osteoporosis, aiming at increase in bone mass. We therefore explored whether Dll4(E12) can be beneficial in female mice if used in combination with PTH. Indeed, hydrodynamic tail vein injection of female mice with *pLIVE-Dll4(E12)* followed by daily PTH1-34 administration over a period of 3 weeks leads to a profound increase in Dll4 immunoreactivity in femoral sections (*Figure 7—figure supplement 1A, B*). This effect is mirrored by significant increases in Col1a1, PDGFRβ, and Emcn immunostaining in response to combined Dll4(E12) and PTH1-34 treatment (*Figure 7—figure supplement 1C-E*). Furthermore, the combination of Dll4(E12) and PTH1-34 increases the abundance of OSX⁺ osteoprogenitors and the length of the metaphysis in adult females, whereas the Opn⁺ area is not significantly enhanced relative to PTH1-34 alone (*Figure 7—figure supplement 2A, B*).

Analysis of Dll4(E12)- and PTH1-34-treated femurs by μ-CT shows that the changes above are accompanied by profound increases in trabecular bone and, in particular, connectivity density and bone surface relative to control females or separate administration of either Dll4(E12) or PTH1-34 (*Figure 7A–G*). In contrast, the beneficial effects of PTH treatment on cortical bone are not enhanced by Dll4(E12) (*Figure 7H–K*). These findings indicate that Dll4(E12) can enhance trabecular bone formation in response to anabolic therapy.

## Analysis of Dll4(E12)-induced processes at single cell resolution

Next, we analyzed long bones from *pLIVE-Dll4(E12)* and control-injected males at 3 wpi by scRNA-seq in order to identify the stromal cell populations mediating the response to Dll4(E12). Following tissue dissociation, hematopoietic cells were depleted from single cell suspensions with magnetic-activated cell sorting and the remaining stromal single cells were captured and barcoded with the BD Rhapsody Express Single-Cell Analysis System (*Figure 8A*). We sequenced approximately 20,000 cells from long bones of four *pLIVE-Dll4(E12)* and control-treated mice. The resulting cell libraries were filtered to remove barcode artifacts and low-quality cells using stringent parameters for low library complexity, mitochondrial gene overrepresentation, cell doublet probability, and low gene expression to ensure that downstream analysis was not affected by technical artifacts (*Figure 8—figure supplement 1A-D*).

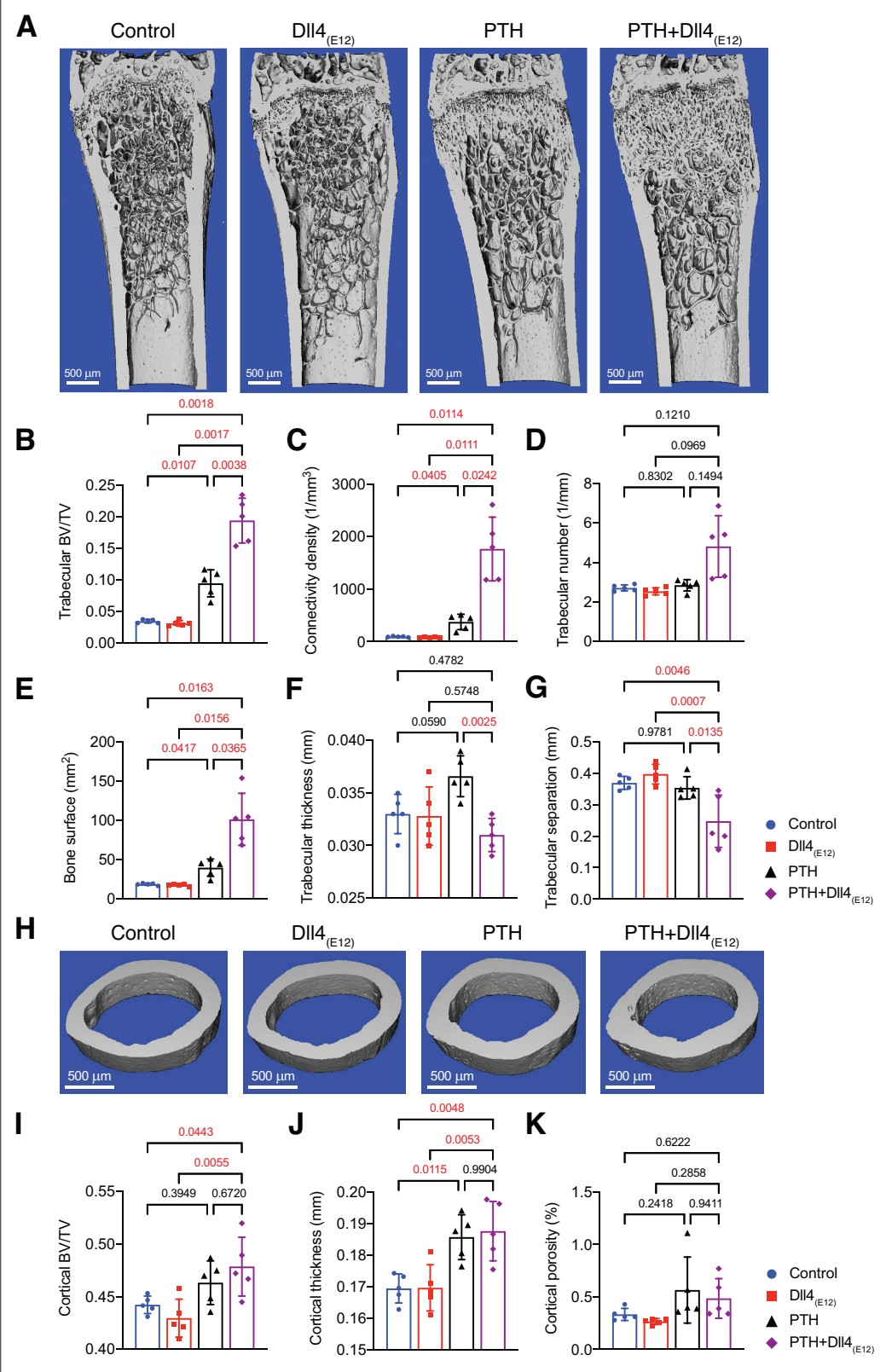

**Figure 7.** Synergistic action of recombinant Dll4(E12) and parathyroid hormone (PTH). (**A**) Representative micro-computed tomography (μ-CT) images of trabecular bone in the distal femoral metaphysis of female wild-type mice treated with vehicle, *pLIVE-Dll4(E12)*, PTH, or the combination of both for 3 weeks. (**B–G**) Quantitative analysis of μ-CT data on trabecular bone volume/total volume (**B**), trabecular connectivity density (**C**), trabecular

*Figure 7 continued on next page*

*Figure 7 continued*

number (**D**), bone surface (**E**), trabecular thickness (**F**), trabecular separation (**G**) in the distal femoral metaphysis. Data represent mean ± SD. (n = 5 mice) (p-values determined by one-way ANOVAs (**F, G**) followed by Sidak's multiple comparisons test or Brown-Forsythe and Welch's ANOVAs (**B, C, D, E**) followed by Dunnett's T3 multiple comparisons test). (**H**) Representative μ-CT images of cortical bone in the female midfemoral diaphysis. (**I–K**) Graphs from μ-CT analysis of the cortical bone volume/total volume (**I**), cortical thickness (**J**), and cortical porosity (**K**) in the midfemoral diaphysis. Data represent mean ± SD. (n = 5 mice) (p-values determined using one-way ANOVAs followed by Sidak's multiple comparisons test).

The online version of this article includes the following source data and figure supplement(s) for figure 7:

**Source data 1.** Source data for *Figure 7B–G,I-K*.

**Figure supplement 1.** Effects of combined parathyroid hormone (PTH) and Dll4$_{(E12)}$ administration.

**Figure supplement 1—source data 1.** Source data for *Figure 7—figure supplement 1B-E*.

**Figure supplement 2.** Synergistic effects of Dll4$_{(E12)}$ and parathyroid hormone (PTH) on osteogenesis.

**Figure supplement 2—source data 1.** Source data for *Figure 7—figure supplement 2A,B*.

A total of 7750 Dll4$_{(E12)}$ cells and 9265 control cells passed filtering and were carried forward for further analysis. To investigate the cell diversity in the Dll4$_{(E12)}$ and control group, we performed clustering of cells based on scaled expression profiles (Louvain algorithm), combined with manual clustering annotation using known and de novo identified marker genes. We identified five distinct cell populations in both *pLIVE-Dll4$_{(E12)}$* and control-injected bone: MSCs, chondrocytes, ECs, osteoblasts, and smooth muscle cells (SMCs) (*Figure 8B–E*). For each population, we then identified all genes that were differentially expressed between the *pLIVE-Dll4$_{(E12)}$* and control samples (Wilcoxon rank sum test). Due to their small number, SMCs were omitted from this analysis. Data analysis reveals that EC and osteoblast populations show no or only minor alterations in response to Dll4$_{(E12)}$. Consistent with this finding, expression of EC markers (*Pecam1*), regulators of vascular growth (*Vegfr2/Kdr*), and of venous (*Nr2f2*, *Vegfr3/Flt4*) or arterial specification (*Sox17*, *Efnb2*) are not altered in the Dll4$_{(E12)}$ group (*Figure 8F,I*). Expression of Notch pathway ligands, receptors, and downstream targets is also only marginally affected in endothelium of the *pLIVE-Dll4$_{(E12)}$*-injected mice (*Figure 8—figure supplement 2A, B*).

Notably, MSCs and chondrocytes show the strongest response to overexpressed Dll4$_{(E12)}$, although chondrocytes display a comparably smaller number of differentially expressed genes (*Figure 8G and H*). Dll4$_{(E12)}$ chondrocytes show normal expression of critical transcripts including the transcription factor Sox9 and the proteoglycan core protein Aggrecan (*Acan*) (*Figure 8J*), consistent with the results from the immunostaining of bone sections (*Figure 3—figure supplement 2C*). In contrast, the alpha1 subunit of type X collagen (*Col10a1*), which is mainly expressed by hypertrophic chondrocytes in the growth plate, is upregulated after Dll4$_{(E12)}$ treatment (p = 3.97e-32) (*Figure 8J*).

In the MSC population, representative markers such as transcripts for the receptor tyrosine kinase PDGFRβ (*Pdgfrb*), a critical regulator of MSC proliferation, and Leptin receptor (*Lepr*), a marker of BM stromal cells, are not altered (*Figure 8K*). However, known pro-osteogenic genes like *Igf2* encoding insulin-like growth factor 2 (*Chen et al., 2010*) are upregulated (p = 9.52e-121), whereas transcripts associated with negative regulation of osteogenesis are downregulated. The latter includes the transcript for insulin-like growth factor-binding protein 3 (*Igfbp3*) (p = 2.80e-166), a secreted protein that binds insulin-like growth factors and limits their bioavailability (*Chen et al., 2010*), and matrix Gla (γ-carboxyglutamate) protein (*Mgp*) (p = 1.11e-159), a negative regulator of calcification (*Yagami et al., 1999*; *Figure 8H* and *Figure 8—figure supplement 2C*). MSCs from Dll4$_{(E12)}$-treated mice showed also increased expression of *Angptl4* (p = 9.82e-132), which encodes angiopoietin-like 4, a positive regulator of osteogenesis that is highly expressed at bone fracture sites but is also a stimulator of osteoclast-mediated bone resorption (*Wilson et al., 2015*; *Knowles et al., 2010*; *Figure 8H*, *Figure 8—figure supplement 2C*). Transcripts for the proteoglycan PRG4, a positive regulator of skeletogenesis and parathyroid hormone-mediated bone formation (*Novince et al., 2012*), are also elevated in Dll4$_{(E12)}$ MSCs (p = 8.39e-112) (*Figure 8H* and *Figure 8—figure supplement 2C*).

GO analyses (hypergeometric test) of each cell population reveals an enrichment of GO terms for morphogenesis, development, vascular growth, and ossification among differentially expressed genes in the MSC population and, to much smaller extent, in chondrocytes (*Figure 8—figure supplement*

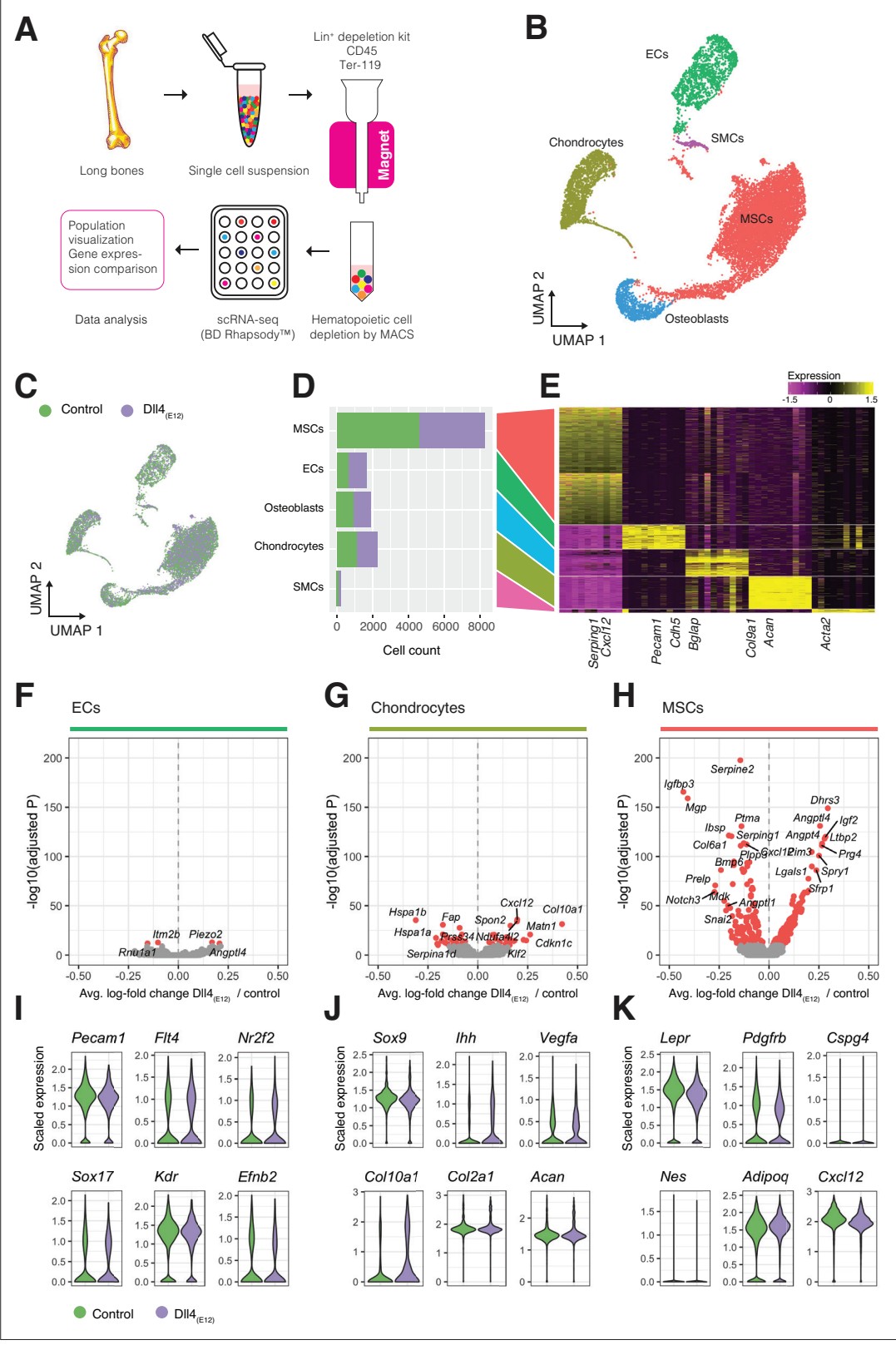

**Figure 8.** Single cell RNA sequencing (scRNA-seq) analysis of control and Dll4$_{(E12)}$-treated bone. (**A**) Overview of the sample processing and scRNA-seq procedure. (**B**) UMAP projection of all cells in *pLIVE-Dll4$_{(E12)}$* and control, colored by Louvain clusters. Endothelial cells (ECs), mesenchymal stromal cells (MSCs), smooth muscle cells (SMCs), chondrocytes, and osteoblasts are indicated. (**C**) UMAP projection of all cells colored by experimental

*Figure 8 continued on next page*

*Figure 8 continued*

condition (green = control, purple = Dll4$_{(E12)}$). (**D, E**) Barplot showing absolute numbers of cells (**D**) and scaled expression heatmap of the top 10 marker genes (**E**) for each of the clusters shown in (**B**). (**F–H**) Differential expression volcano plots showing -log10(adjusted p-value) against average log-fold change of *pLIVE-Dll4$_{(E12)}$* vs. control in ECs (**F**), chondrocytes (**G**), and MSCs (**H**). Genes with adjusted p-values smaller than 1e-10 are colored in red. (**I–K**) Selected cell population relevant gene expression shown as violin (density) plots for EC (**I**), chondrocyte, (**J**) and MSC (**K**) subpopulations.

The online version of this article includes the following source data and figure supplement(s) for figure 8:

**Source data 1.** Source data for *Figure 8F–H*.

**Figure supplement 1.** Single cell RNA sequencing (scRNA) quality control statistics.

**Figure supplement 2.** Extended analysis of differentially expressed genes.

**Figure supplement 2—source data 1.** Source data for *Figure 8—figure supplement 2A-E*.

---

*2D*). Analysis of endogenous Notch receptor (*Notch1-4*) and ligand (*Dll1, Dll4 and Jag1, Jag2*) expression in Dll4$_{(E12)}$-treated stromal cells, however, indicates no alterations except the downregulation of *Notch3* in MSCs (p = 1.10e-64) (*Figure 8—figure supplement 2A, B*). While Dll4$_{(E12)}$ results in a strong increase of OSX$^+$ cells in vivo, scRNA-seq analysis reveals very limited gene expression changes in osteoblasts (*Figure 8—figure supplement 2E*). Together, these data indicate that the pro-osteogenic effects of Dll4$_{(E12)}$ are mediated by MSCs.

## Dll4$_{(E12)}$ effects within the bone MSC subpopulations

Bone MSCs are highly heterogeneous, found in different locations and can be subdivided based on marker gene expression (*Baryawno et al., 2019*; *Pinho and Frenette, 2019*; *Zhou et al., 2014*; *Mizoguchi et al., 2014*). Subclustering (Louvain algorithm) of the MSC population in our scRNA-seq data identifies three distinct subpopulations (*Figure 9A–C*). Diaphyseal MSCs (dpMSCs) show high expression of *Lepr* and *Kitl*, the gene encoding stem cell factor, the ligand of c-Kit receptor (*Figure 9A–C and E*). Metaphyseal MSCs (mpMSCs) characterized by expression of markers associated with osteogenesis, namely of the transcription factor OSX (encoded by the gene *Sp7*), the secreted extracellular matrix protein periostin (*Postn*), and the transcriptional repressor and Notch pathway gene *Hey1* (*Figure 9A–C and E*). In addition, we find a small population of fibroblast-like cells, which express the proteoglycan Decorin (*Dcn*) and the glycosyl phosphatidylinositol-anchored cell surface protein Sca-1 (*Ly6a*) (*Figure 9A–C*). Notably, dpMSCs represent 85–86% of total MSCs in both the Dll4$_{(E12)}$ and control group (*Figure 9A and B*). Most of genes that are differentially expressed in Dll4$_{(E12)}$-treated samples are also found in the dpMSC subpopulation. This includes the upregulated genes *Igf2* (p = 5.39e-88) and *Prg4* (p = 3.05e-90), which are known to promote osteogenesis, and the downregulated genes *Igfbp3* (p = 1.69e-156) and *Mgp* (p = 9.76e-148), which are inhibitors of bone formation (*Figure 9D and E*). Furthermore, UMAP projections show a significant increase in the fraction of *Sp7/OSX*-expressing cells in the dpMSC population (*Figure 9F*). Consistent with the *Hey1-EGFP* reporter results (*Figure 3—figure supplement 1A*), *Hey1* expression labels mpMSCs and a small fraction of dpMSCs. The latter might, as the *Tg(Hey1-EGFP)$^{ID40Gsat}$* reporter expression suggests (*Figure 3—figure supplement 1A*), represent cells in the transition zone between the metaphysis and diaphysis.

The scRNA-seq data show no significant increase in total *Hey1$^+$* MSCs, which is presumably due to the low expression and therefore limited detection of this transcript (*Figure 9F*). Taken together, our results support that Dll4$_{(E12)}$ treatment promotes osteogenesis through effects in MSCs and, in particular, the dpMSC population. In contrast, molecular changes are limited in the EC and osteoblast populations, suggesting that these cells are not exposed or unresponsive to Dll4$_{(E12)}$.

## Discussion

Notch signaling has multiple important functional roles in the skeletal system, including the maintenance and expansion of the mesenchymal progenitor cell pool and the coupling of angiogenesis and osteogenesis (*Ramasamy et al., 2014*; *Hilton et al., 2008*; *Engin et al., 2008*; *Kusumbe et al., 2014*; *Golson et al., 2009*; *Limbourg et al., 2005*). Interactions between Dll4 and Notch receptors,

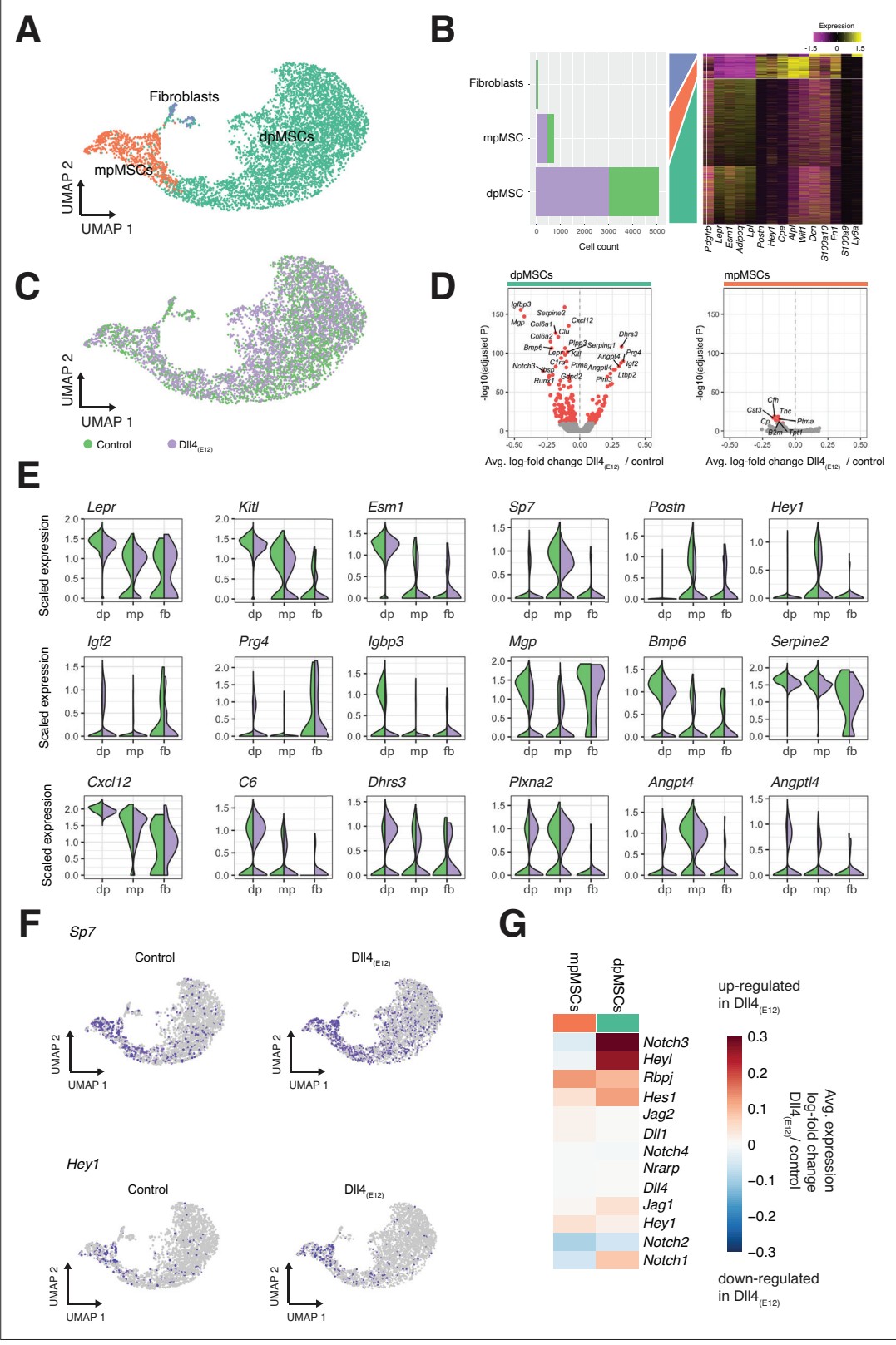

**Figure 9.** Single cell RNA sequencing (scRNA-seq) analysis of mesenchymal stromal cell (MSC) subclusters.
(**A**) UMAP projection of MSCs in both control and *pLIVE-Dll4(E12)* samples colored by Louvain clusters. Metaphyseal
MSCs (mpMSCs), diaphyseal MSCs (dpMSCs), and fibroblasts are indicated. (**B**) Barplot showing absolute numbers
of cells per sample (left), and scaled expression heatmap of the top 10 marker genes (right) in each of the clusters

*Figure 9 continued on next page*

*Figure 9 continued*

shown in (**A**). (**C**) UMAP projection of all cells colored by experimental condition green = control, purple = Dll4$_{(E12)}$. (**D**) Differential expression volcano plots showing -log10(adjusted p-value) against average log-fold change of treatment/control for dpMSCs (left) and mpMSCs (right). Genes with an adjusted p-value less than 1e-10 are colored in red. (**E**) Violin plots showing selected differentially expressed genes in the three MSC subpopulations. Each violin is split along the vertical axis into control and *pLIVE-Dll4*$_{(E12)}$. (**F**) UMAP projections of control and Dll4$_{(E12)}$-treated cells colored for the expression of *Sp7* and *Hey1*. (**G**) Heatmap of the average expression log-fold change of selected Notch pathway genes with a p-value < 1e-10 and log-fold change >0.2 in dpMSCs and mpMSCs.

The online version of this article includes the following source data for figure 9:

**Source data 1.** Source data for *Figure 9D–G*.

predominantly Notch1 but also Notch4, negatively control EC proliferation and angiogenic growth of the vasculature in most organs. In bone, however, Notch signaling promotes angiogenesis and triggers the expansion of type H vessels, which, in turn, results in increased osteogenesis (*Ramasamy et al., 2014*; *Ramasamy et al., 2016*). To promote bone angiogenesis and osteogenesis, we have designed synthetic Notch ligands containing poly-Asp peptide motifs, which have been previously shown to bind different regions in long bone including the primary spongiosa in proximity of the growth plate (*Kasugai et al., 2000*; *Miller et al., 2008*), a region containing a high amount of angiogenic vessels. It has also been shown that an Asp$_{6x}$ peptide preferentially binds to areas of bone resorption, whereas a different motif, (AspSerSer)$_{6x}$, targets bone-forming surfaces (*Zhang et al., 2012*). Future work will have to address whether other bone surface-binding moieties will endow Notch ligands with distinct and perhaps even more potent biological properties due to the activation of different target cell populations. Our scRNA-seq analysis shows, for example, that the Asp$_{8x}$ motif in Dll4$_{(E12)}$ leads to very limited changes in bone ECs. This might reflect that the spatial distance between the bone surface and ECs precludes sufficient physical contact between hydroxyapatite-bound Dll4$_{(E12)}$ and endothelial Notch receptors.

Nevertheless, we found that Dll4$_{(E12)}$ can trigger an increase in trabecular bone, whereas similar effects were not seen for Dll4-Asp$_{8x}$, Jag1-Asp$_{8x}$, or Jag1$_{(JV1)}$. The underlying reasons for this difference are not clear, but it is possible that only Dll4$_{(E12)}$ accomplishes a critical level of Notch activation required for bone formation in adult male mice. This should, however, not lead to conclusions about the functional roles of endogenous Notch ligand expression. In fact, numerous reports have shown that Jagged1 can regulate osteogenic differentiation of bone MSCs and the ligand is also highly upregulated during fracture repair (*Li et al., 2009*; *Hill et al., 2014*; *Dishowitz et al., 2014*; *Osathanon et al., 2013*).

Notch signaling was shown to inhibit chondrocyte differentiation by suppressing the expression of the transcription factor Sox9, a known regulator of chondrogenesis (*Chen et al., 2013*). Genetic disruption of Notch signaling in embryonic limb bud mesenchyme using *Prx1-Cre* transgenic mice led to the accumulation of hypertrophic chondrocytes in the growth plate, whereas chondrocyte proliferation was strongly reduced (*Hilton et al., 2008*). Our findings show that Dll4$_{(E12)}$ treatment of adult mice increases the expression of *Col10a1* in chondrocytes, which encodes the alpha chain of type X collagen, a short chain collagen expressed by hypertrophic chondrocytes that is essential for growth plate development and mineralization of trabecular bone (*Kwan et al., 1997*). The absence of appreciable accumulation of Dll4$_{(E12)}$ inside the growth plate, however, suggests that these alterations might be indirect. It was also shown that conditional activation of Notch signaling in osteocytes increases bone formation and mineralization, which is sufficient for the rescue of both age-associated and ovariectomy-induced bone loss (*Liu et al., 2016*).While our work does not exclude anabolic effects of Dll4$_{(E12)}$ through the activation of Notch signaling in osteocytes, these cells were not recovered in our scRNA-seq data presumably due to the conditions of tissue dissociation.

Instead, our results indicate that Dll4$_{(E12)}$ activates Notch signaling primarily in immature MSCs and leads to the expansion of this population and, as shown previously (*Hilton et al., 2008*), reduced Runx2 expression. It is also fully consistent with previous findings showing that genetic approaches leading to Notch activation in mesenchymal progenitors suppress differentiation and promote the expansion of these cells (*Hilton et al., 2008*; *Engin et al., 2008*; *Canalis et al., 2013*). In this context, it is presumably highly advantageous that Dll4$_{(E12)}$-mediated Notch activation is, in contrast to irreversible genetic

modifications, transient and does not suppress osteoblastic differentiation, presumably because cells are not permanently exposed to the bone surface-bound recombinant protein. This is indicated by the absence of *Hey1-GFP* signal in OSX[+] cells but also the absence of major transcriptional alterations in osteoblasts.

MSCs are a heterogenous population of stromal cells, which includes cells with progenitor properties, colony-forming capacity ex vivo, and the potential to generate bone, fat, fibroblasts, and other cell types (*Zhou et al., 2014*; *Bianco and Gehron Robey, 2000*; *Nombela-Arrieta et al., 2011*; *Ortinau et al., 2019*). MSCs are likely to include comparably rare mesenchymal stem and progenitor cells characterized by the ability to give rise to multiple differentiated cell types in a clonal fashion (*Bianco and Gehron Robey, 2000*; *Nombela-Arrieta et al., 2011*; *Ono et al., 2014*; *Uccelli et al., 2008*). It might appear surprising that our scRNA-seq data show the strongest gene expression changes in dpMSCs, but previous work has established that vessel-associated MSCs in the diaphysis can give rise to bone-forming cells (*Zhou et al., 2014*; *Mizoguchi et al., 2014*; *Sivaraj et al., 2021*). In fact, expansion of the metaphyseal region in response to Dll4$_{(E12)}$ might well involve the incorporation of dpMSCs from the adjacent transition zone and marrow. Taken together, we propose that the pro-osteogenic capacity of Dll4$_{(E12)}$ is primarily mediated by Notch-controlled expansion of immature MSCs. It is also feasible that bone formation is enhanced through the release of secreted signaling molecules acting in a paracrine fashion, which might apply to the alterations seen in chondrocytes.

Our study not only proves that the administration of exogenous Notch ligand can induce bone formation in adult mice, the results also show that this treatment does not lead to adverse side effects. Circulating Dll4$_{(E12)}$ might act as an antagonist and therefore disrupt critical endogenous Notch-ligand interactions in multiple organs, which might lead to defects in small intestine, liver, or T cell development that are known to be caused by Notch inhibition (*Yan et al., 2010*; *Radtke et al., 1999*; *van Es et al., 2005*). The absence of such defects might reflect efficient retention of Dll4$_{(E12)}$ fusion protein in bone and thereby low bioavailability in other organs. It is, nevertheless, remarkable that even liver, the site of Dll4$_{(E12)}$ expression, does not show overt morphological alterations.

Together, our findings establish that bone-targeting and binding-optimized Dll4 ligand can be used to stimulate osteogenesis in adult mice without adverse side effects. We propose that Dll4$_{(E12)}$ might serve as a promising example for the design of future anti-osteoporosis drugs.

# Materials and methods

## Key resources table

| Reagent type (species) or resource | Designation | Source or reference | Identifiers | Additional information |
|---|---|---|---|---|
| Strain, strain background (*Mus musculus*, C57BL/6JRj) | WT | Janvier Labs | | |
| Genetic reagent (*Mus musculus*) | *Tg(Hey1-EGFP)$^{ID40Gsat}$* | GENSAT | MGI:4847129 | |
| Recombinant DNA reagent | pcDNA3.1- Dll4(ECD)- His(6x)-Asp(8x) (plasmid) | This paper | | Dll4-Asp$_{(8x)}$ |
| Recombinant DNA reagent | pcDNA3.1- Dll4(ECD)- Variant- His(6x)-Asp(8x) (plasmid) | This paper | | Dll4$_{(E12)}$ |
| Recombinant DNA reagent | pLIVE-Dll4(ECD)-His(6x)-Asp(8x) (plasmid) | This paper | | *pLIVE-Dll4-Asp(8x)* |
| Recombinant DNA reagent | pLIVE-Dll4(ECD) Variant- His(6x)-Asp(8x) (plasmid) | This paper | | *pLIVE-Dll4(E12)* |
| Recombinant DNA reagent | pcDNA3.1- Jag1 (ECD)- His(6x)-Asp(8x) (plasmid) | This paper | | Jag1- Asp$_{(8x)}$ |
| Recombinant DNA reagent | pcDNA3.1- Jag1 (ECD)- JV1- His(6x)-Asp(8x) (plasmid) | This paper | | Jag1$_{(JV1)}$ |
| Recombinant DNA reagent | pLIVE- Jag1 (ECD)- His(6x)-Asp(8x) (plasmid) | This paper | | *pLIVE-Jag1- Asp(8x)* |
| Recombinant DNA reagent | pLIVE- Jag1 (ECD)- JV1- His(6x)-Asp(8x) (plasmid) | This paper | | *pLIVE-Jag1(JV1)* |

*Continued on next page*

*Continued*

| Reagent type (species) or resource | Designation | Source or reference | Identifiers | Additional information |
|---|---|---|---|---|
| Cell line (*Homo sapiens*) | Human Umbilical Vein Endothelial Cells (HUVEC) | ThermoFisher | Cat# C0035C | Cell identity and absence of mycoplasma contamination or human pathogens were certified by the supplier |
| Cell line (*Homo sapiens*) | Human Embryonal kidney –293 (HEK293) | DSMZ | Cat# ACC 305 | Cell identity and absence of mycoplasma contamination were certified by the supplier |
| Antibody | Anti-Endomucin (Rat, monoclonal) | Santa Cruz | Cat# SC-65495 | IF (1:100) |
| Antibody | Anti-PECAM-1 (Rat, monoclonal) | Pharmigen | Cat# 553,370 | IF (1:100) |
| Antibody | Anti-CD31 (Goat, polyclonal) | R&D Systems | Cat# FAB3628 | IF (1:100) |
| Antibody | Anti-Pdgfrβ (Goat, polyclonal) | R&D Systems | Cat# AF1042 | IF (1:100) |
| Antibody | Anti-NG2 (Rabbit, polyclonal) | Millipore | Cat# AB5320 | IF (1:100) |
| Antibody | Anti-BCAM (Goat, polyclonal) | R&D Systems | Cat# AF8299 | IF (1:50) |
| Antibody | Anti-ATP6V1B1+ ATP6V1B2 (Rabbit, polyclonal) | Abcam | Cat# ab200839 | IF (1:100) |
| Antibody | Anti-Aggrecan (Rabbit, polyclonal) | Millipore | Cat# AB1031 | IF (1:100) |
| Antibody | Anti-Sox9 (Goat, polyclonal) | R&D Systems | Cat# AF3075 | IF (1:100) |
| Antibody | Anti-Perilipin (Rabbit, polyclonal) | Cell Signaling | Cat# 9349 | IF (1:100) |
| Antibody | Anti-Osterix (Rabbit, polyclonal) | Abcam | Cat# ab22552 | IF (1:1000) |
| Antibody | Anti-Collagen Type I (Rabbit, polyclonal) | Millipore | Cat# AB765P | IF (1:200) |
| Antibody | Anti-Osteopontin (Goat, polyclonal) | R&D Systems | Cat# AF808 | IF (1:100) |
| Antibody | Anti-Osteocalcin (Rabbit, polyclonal) | LifeSpan BioSciences | Cat# LS-C17044 | IF (1:100) |
| Antibody | Anti-Runx2 (Rabbit, monoclonal) | R&D Systems | Cat# MAB2006 | IF (1:50) |
| Antibody | Anti-Dll4 (Goat, polyclonal) | R&D Systems | Cat# AF1389 | IF (1:100) WB (1:200) |
| Antibody | Anti-Jag1 (Goat, polyclonal) | Sigma | Cat# J4127 | IF (1:100) WB (1:500) |
| Antibody | Anti-GAPDH (Rabbit, monoclonal) | Ambion | Cat# AM4300 | WB (1:1000) |
| Antibody | Anti-CD45-FITC (Rat, monoclonal) | eBioscience | Cat# 11–0451 | |
| Antibody | Anti-CD8-Biotin (Rat, monoclonal) | eBioscience | Cat# 13–0081 | |
| Antibody | Anti-CD4-APC (Rat, monoclonal) | eBioscience | Cat# 17-0042-82 | |
| Antibody | Streptavidin PE/Cy7 | Thermo Scientific | Cat# SA1012 | |
| Antibody | Lineage Cell Depletion Kit | Miltenyi Biotec | Cat# 130-090-858 | |
| Antibody | CD45 Microbeads | Miltenyi Biotec | Cat# 130-052-301 | |
| Antibody | CD117 Microbeads | Miltenyi Biotec | Cat# 130-091-224 | |
| Antibody | Ter-119 Microbeads | Miltenyi Biotec | Cat# 130-049-901 | |
| Antibody | Anti-rabbit Alexa Fluor-488 (Donkey, polyclonal) | Invitrogen | Cat# A21206 | IF (1:500) |
| Antibody | Anti-rabbit Alexa Fluor-594 (Donkey, polyclonal) | Invitrogen | Cat# A21207 | IF (1:500) |
| Antibody | Anti-rabbit Alexa Fluor-647 (Donkey, polyclonal) | Invitrogen | Cat# A31573 | IF (1:500) |
| Antibody | Anti-rat Alexa Fluor-488 (Donkey, polyclonal) | Invitrogen | Cat# A21208 | IF (1:500) |

*Continued on next page*

*Continued*

| Reagent type (species) or resource | Designation | Source or reference | Identifiers | Additional information |
|---|---|---|---|---|
| Antibody | Anti-rat Alexa Fluor-594 (Donkey, polyclonal) | Invitrogen | Cat# A21209 | IF (1:500) |
| Antibody | Anti-rat Alexa Fluor-647 (Donkey, polyclonal) | Jackson ImmunoResearch | Cat# 712-605-153 | IF (1:500) |
| Antibody | Anti-rat Alexa Fluor-647 (Donkey, polyclonal) | Invitrogen | Cat# A21247 | IF (1:500) |
| Antibody | Anti-goat Alexa Fluor-488 (Donkey, polyclonal) | Invitrogen | Cat# A11055 | IF (1:500) |
| Antibody | Anti-goat Alexa Fluor-647 (Donkey, polyclonal) | Invitrogen | Cat# A21447 | IF (1:500) |
| Antibody | Anti-goat Alexa Fluor-647 (Donkey, polyclonal) | Thermo Scientific | Cat# A32849 | IF (1:500) |
| Antibody | Anti-rabbit IgG HRP-linked (Goat) | Cell Signaling | Cat# 7074 | WB (1:15000) |
| Antibody | Anti-goat IgG HRP-linked (Donkey) | Antibody online | Cat# ABIN1536502 | WB (1:15000) |
| Antibody | Anti-mouse IgG 656G HRP-linked (Sheep) | GE Healthcare | Cat# NA931 | WB (1:40000) |
| Antibody | Anti-goat IgG (H + L) Peroxidase AffiniPure Bovine | Jackson ImmunoResearch | Cat# 805-035-180 | WB (1:15000) |
| Sequence-based reagent | Human *GAPDH* Endogenous Control (VIC/MGB probe, primer limited) | Applied Biosystems | Cat# 4326317E | TaqMan probe Hs99999905_m1 |
| Sequence-based reagent | Human *ACTB* Endogenous Control (VIC/MGB probe, primer limited) | Applied Biosystems | Cat# 4326315E | TaqMan probe Hs99999903_m1 |
| Sequence-based reagent | Human *HEY1* TaqMan Gene Expression Assay (FAM) | Applied Biosystems | Cat# 4331182 | TaqMan probe Hs00232618_m1 |
| Sequence-based reagent | Human *DLL4* TaqMan Gene Expression Assay (FAM) | Applied Biosystems | Cat# 4331182 | TaqMan probe Hs00184092_m1 |
| Sequence-based reagent | Human *HEY2* TaqMan Gene Expression Assay (FAM) | Applied Biosystems | Cat# 4331182 | TaqMan probe Hs00232622_m1 |
| Sequence-based reagent | Human *HES1* TaqMan Gene Expression Assay (FAM) | Applied Biosystems | Cat# 4331182 | TaqMan probe Hs00172878_m1 |
| Sequence-based reagent | Human *EFNB2* TaqMan Gene Expression Assay (FAM) | Applied Biosystems | Cat# 4331182 | TaqMan probe Hs00187950_m1 |
| Sequence-based reagent | Human *NRARP* TaqMan Gene Expression Assay (FAM) | Applied Biosystems | Cat# 4331182 | TaqMan probe Hs01104102_s1 |
| Peptide, recombinant protein | Recombinant Human PTH (1-34) | BACHEM | Cat# H-4835-GMP, 4033364 | |
| Commercial assay or kit | BCA Protein Assay Kit | Pierce | Cat# 23225 | |
| Commercial assay or kit | RNeasy Plus Micro Kit | QIAGEN | Cat# 74034 | |
| Commercial assay or kit | iScript cDNA Synthesis Kit | BIO-RAD | Cat# 170–8891 | |
| Commercial assay or kit | SsoAdvanced Universal Probes Supermix | BIO-RAD | Cat# 172–5284 | |
| Commercial assay or kit | LEGENDplex Mouse Inflammation Panel (13-plex) with V-bottom plates | BioLegend | Cat# 740446 | |
| Commercial assay or kit | Anticoagulant EDTA-treated Microvettes | Sarstedt | Cat# 20.1341 | |
| Commercial assay or kit | BD Rhapsody Whole Transcriptome Analysis (WTA) Amplification kit | BD Biosciences | Cat# 633,801 | |
| Commercial assay or kit | Agencourt AMPure XP magnetic beads | Beckman Coulter Life Sciences | Cat# A638880 | |
| Commercial assay or kit | Disposable polystyrene columns | Thermo Scientific | Cat# 29922 | |
| Commercial assay or kit | Amicon Ultra-0.5 Centrifugal Filter | Millipore | Cat# UFC500396 | |

*Continued*

| Reagent type (species) or resource | Designation | Source or reference | Identifiers | Additional information |
|---|---|---|---|---|
| Commercial assay or kit | Pierce Slide-A-Lyzer 10K MWCO Dialysis Cassettes | Thermo Scientific | Cat# 66380 | |
| Chemical compound, drug | Ni-NTA agarose resin | Qiagen | Cat# 302010 | |
| Chemical compound, drug | Sucrose | Sigma | Cat# S0389 | |
| Chemical compound, drug | cOmplete ULTRA Tablets Protease Inhibitor Cocktail | Roche | Cat# 05892970001 | |
| Chemical compound, drug | Phosphatase inhibitor cocktail set V | EMD Millipore | Cat# 524629 | |
| Chemical compound, drug | SimplyBlue SafeStain | Invitrogen | Cat# LC6060 | |
| Chemical compound, drug | Gelatine | Sigma | Cat# G1890 | |
| Chemical compound, drug | Polyvinylpyrrolidone | Sigma | Cat# P5288 | |
| Chemical compound, drug | Trypsin-EDTA solution | Sigma | Cat# T3924 | |
| Chemical compound, drug | Paraformaldehyde | Sigma | Cat# P6148 | |
| Chemical compound, drug | 70 kDa Dextran, Texas Red Lysine fixable | Molecular Probes | Cat# D1864 | |
| Chemical compound, drug | Calcein | Sigma | Cat# C0875 | |
| Chemical compound, drug | Fluoromount-G | Southern Biotech | Cat# 0100–01 | |
| Chemical compound, drug | ECL Prime Western Blotting Detection Reagent | GE-Healthcare | Cat# RPN2236 | |
| Chemical compound, drug | 4-Hydroxy tamoxifen | Sigma | Cat# H7904 | |
| Chemical compound, drug | Hematoxilin | Sigma | Cat# MHS16 | |
| Chemical compound, drug | Dimethyl sulfoxide | Sigma | Cat# D8418 | |
| Chemical compound, drug | MEM200 endothelial cells medium | ThermoFisher | Cat# M200500 | |
| Chemical compound, drug | LSGS | ThermoFisher | Cat# S00310 | |
| Chemical compound, drug | EBM-2 endothelial cells medium | Lonza | Cat# CC-3156 | |
| Chemical compound, drug | EGM-2 Single Quots | Lonza | Cat# CC-4176 | |
| Chemical compound, drug | Opti-MEM | Gibco | Cat# 31985–047 | |
| Chemical compound, drug | Poly-L-lysine | Sigma | Cat# P6282 | |
| Chemical compound, drug | HEPES | Sigma | Cat# H3537 | |
| Chemical compound, drug | Ketamine | Zoetis | Cat# 344771 | |
| Chemical compound, drug | Rompum | Bayer HealthCare | Cat# D-51368 | |
| Chemical compound, drug | Collagenase I | Gibco | Cat# 17100–017 | |
| Chemical compound, drug | Collagenase IV | Gibco | Cat# 17104–019 | |
| Chemical compound, drug | Dispase | Gibco | Cat# 17105–041 | |
| Software, algorithm | ImageJ (v2.0.0 Fiji) | *Schindelin et al., 2012* | | |
| Software, algorithm | Volocity (v6.3) | Perkin Elmer | | |
| Software, algorithm | Illustrator (vCC2018) | Adobe | | |
| Software, algorithm | GraphPad Prism7 | GraphPad Software | | |
| Software, algorithm | FlowJo (v10.3) | BD Life Sciences | | |
| Other | DAPI stain | Sigma | Cat# D9542 | (1 mg/mL) |

## Plasmid construction

To generate the Dll4-Asp$_{8x}$ soluble protein containing the Dll4 ECD fused to 6x His and 8x Asp peptide sequences, a cDNA fragment encoding amino acids 1–530 of murine Dll4 (https://www.uniprot.org/uniprot/Q9JI71) was amplified via PCR using the following oligonucleotide primers:

*Dll4(ECD)-BamHI-Fwd*: 5'-ACTACATATAGGATCCaccctaggatttgctccagg-3' and *Dll4(ECD)-PolyD_XhoI-Rev*: 5'-CTTATGATCTCGAGttagtcgtcgtcgtcgtcgtcgtcgtgatggtgatggtgatgatctgttctgtttttcag aggacgc-3'.

To generate the plasmid encoding the affinity variant Dll4$_{(E12)}$, which contains several point muta-tions in the ECD (*Canalis et al., 2013*), the plasmid encoding Dll4-Asp$_{8x}$ was modified with the QuickChange Lightning Multi Site-Directed Mutagenesis Kit (Cat#201515, Agilent Technologies) and the following primers: *Dll4-g1242a*: 5'-gctgcgggctccagcatcttccagc-3', *Dll4-t1479c*: 5'-cctctgc agttgcccctcaatttcacctggc-3', *Dll4-a1513t*: 5'-ggaaccttctcactcatcatccaagcttggcac-3', *Dll4-a1587t*: 5'-aaactctctcatcagccaatt catcatccaaggctc-3', *Dll4-c1740t*: 5'-gcaagaagcgcgatgactacttcggacattatgag -3', *Dll4-t1777c*: 5'-cagatggcagcccgtcctgcctgcc-3', *Dll4-a1803g*: 5'-ctggtcacagtactccccagtccagcccg -3'.

The generation of the Jag1-Asp$_{8x}$ soluble protein containing the Jag1 ECD fused to 6x His and 8x Asp peptide sequences involved a strategy analogous to the construction of Dll4-Asp$_{8x}$. A cDNA covering amino acid residues 1–1067 of the murine Jagged1 ECD (https://www.uniprot.org/uniprot/Q9QXX0) was amplified with the following oligonucleotide primers: *Jagged1(ECD)-BamHI-Fwd*: 5'-ACTACATATAGGATCCgccgcagcgatgcggtccccacg-3' and *Jagged1(ECD)-XhoI-PolyD-Rev*: 5'-CTTAT GATCTCGAGttagtcgtcgtcgtcgtcgtcgtcgtgatggtgatggtgatgatctgttctgtttttcagaggac gc-3'. Site-directed mutagenesis was used to generate Jag1$_{(JV1)}$, a high affinity version of the Jag1 ECD due to several point mutations (*Luca et al., 2017*), linked to 6x His and 8x Asp peptides with the following primers *Jagged1-c260g&t261g*: 5'-gtatcagtcccgcgtcagggccgggggga-3', *Jagged1-c95t*: 5'-ggtgtgcgg ggccttgggtcagtttg-3', *Jagged1-c202g*: 5'-cgcaagtgcaccggcgacgagtgt g-3', *Jagged1-g214a*: 5'-gcac cGgcgacgagtgtaatacgtacttcaaagtg-3', *Jagged1-a545g&g546a*: 5'-attgcccacttcgagtatcgaatccgagtga cctgtgatg-3'.

All PCRs were carried out using PrimeSTAR Max DNA Polymerase (R045B, Takara). The resulting PCR fragments were digested by restriction enzymes BamHI and XhoI (New England Biolabs) and the resulting fragments were inserted into the in vivo overexpression vector *pLIVE* (MIR-5420, Mirus) for in vivo studies and into the expression vector *pcDNA3.1* (Invitrogen) for experiments in cultured cells.

## HEK293 cell transfection and His-tagged protein purification

For overexpressing Dll4 or Jag1 recombinant proteins, $4 \times 10^6$ HEK293 cells were plated per 10 cm dish. Twenty hours later medium was refreshed and after 3 hr cells in each dish were transfected with 18 µg plasmid DNA of *pcDNA3.1-Dll4-Asp$_{8x}$*, *pcDNA3.1-Dll4$_{(E12)}$*, *pcDNA3.1-FRT-Jag1(ECD)-His$_{(6x)}$-Asp$_{8x}$*, and *pcDNA3.1-FRT-Jag1(ECD)$_{(JV1)}$-His$_{(6x)}$-Asp$_{8x}$* using the CalPhos Mammalian Transfection kit (631312, TaKaRa Bio), following the manufacturer's instructions. Sixteen hours later Opti-MEM culture medium (31985–047, Gibco) was added to the cells.

Culture supernatants were collected at 24 hr post-transfection and clarified by centrifugation at 300× *g* for 15 min at 4°C. For whole cell lysate samples, cells were washed twice with PBS and 1 mM PMSF and incubated with RIPA modified buffer (20 mmol/L Tris-HCl, pH 8.0, 150 mmol/L NaCl, 0.5% Triton X-100, 0.1% SDS, 0.1% Na-DOC, 2 mmol/L EDTA, cOmplete ULTRA (05892970001, Roche), and phosphatase inhibitor cocktail set V (524629, EMD Millipore)) for 20 min at 4°C. Lysates were further sonicated, spun down at 4°C for 10 min at full speed, and protein concentration was quantified using BCA Protein Assay Kit (23225, Pierce).

For the purification of His-tagged recombinant proteins from supernatants, Ni-NTA agarose resin (302010, QIAGEN) packed in disposable polystyrene columns (29922, Thermo Scientific) were used. Binding of recombinant proteins to Ni-NTA resin was done in a buffer containing 20 mmol/L Tris-HCl, 150 mmol/L NaCl, and 4 mmol/L imidazole at pH 7.4. After the passage of lysates, beads were washed with wash buffer containing 20 mmol/L Tris-HCl, 150 mmol/L NaCl, and 20 mmol/L imidazole at pH 7.4 before His-tagged proteins were eluted into elution buffer containing 20 mmol/L Tris-HCl, 150 mmol/L NaCl, and 500 mmol/L imidazole, pH 7.4.

To remove imidazole, eluted proteins were dialyzed using Pierce Slide-A-Lyzer 10 K MWCO Dialysis Cassettes (66380, Thermo Scientific) in dialysis buffer (20 mmol/L Tris-HCl, 150 mmol/L NaCl at pH 7.4). To adjust the loading volume, 500 µL of supernatant, flow-through, and wash solution were concentrated to 30 µL using Amicon Ultra-0.5 Centrifugal Filter Unit with 3 KDa cutoff (UFC500396, Millipore). After quantitation, whole cell lysates, concentrated protein solutions, and eluted recom-binant proteins were mixed with SDS loading buffer and boiled for 10 min. Samples were run on

8% SDS-PAGE gel and visualized by SimplyBlue SafeStain (LC6060, Invitrogen). Target proteins were detected by comparison with protein standard markers.

## HUVEC cell culture and Dll4 or Jag1 treatment in vitro

For Dll4 experiments, HUVECs (ThermoFisher, C0035C) were cultured in MEM200 medium (M200500, ThermoFisher) supplemented with LSGS (S00310, ThermoFisher), 2% FCS (631106, Clontech), and 1% penicillin-streptomycin solution (P4333, SIGMA). For Jag1 experiments, HUVECs (ThermoFisher, C0035C) were cultured in EBM2 medium (33156, Lonza) supplemented with EGM2 bullet kit (CC-4176, Lonza). HUVECs were grown in a humidified atmosphere with 5% $CO_2$ at 37°C. Cells were tested for mycoplasma contamination by the supplier. For Notch activation assays with immobilized Dll4 or Jag1 recombinant proteins, poly-L-lysine (P6282, SIGMA) coating was used in combination with Dll4-Asp$_{8x}$, Dll4$_{(E12)}$, Jag1-Asp$_{8x}$, or Jag1$_{(JV1)}$. Corning 60 mm × 15 mm culture dishes (430196, Corning) were pre-incubated with 0.2 mg of poly-L-lysine in 0.1 M sodium borate buffer (pH = 8.5) overnight at 37°C in the cell culture incubator (Heracell 240i, ThermoFisher). Dishes were washed four times with $H_2O$ followed by four times additional washes with PBS. Following incubation with 4 µg of Dll4, Jag1 recombinant proteins or BSA in 2 mL PBS for 12 hr at 4°C and four washes with PBS, each culture dish was seeded with $2.2 \times 10^5$ cells 4 mL complete medium and cultured for 16 hr.

For Notch inhibition assays, $4.5 \times 10^5$ cells per well were cultured in Corning six-well plates pre-coated with 0.1% Gelatin. The final volume of M200+ LSGS or EGM2 medium was adjusted to 2 mL and 20 µg of Dll4 recombinant, 35 µg of Jag1 recombinant proteins or BSA protein was added to confluent HUVECs for 8 hr at 37°C.

## Quantitative RT-PCR

For the analysis of gene expression, total RNA from HUVECs was collected directly from the culture dish with 350 µL RLT plus lysis buffer (1053393, Qiagen) followed by immediate RNA extraction using RNeasy Plus Micro Kit (Qiagen, 74034). Complementary DNA was generated with 500 ng RNA per reaction using iScript cDNA Synthesis Kit (170–8890, BIO-RAD). Quantitative PCR (qPCR) was performed on a CFX96 Touch Real-Time PCR Detection System (BIO-RAD). The following FAM-conjugated TaqMan gene expression probes (ThermoFisher) were used in combination with SsoAdvanced Universal Probes Supermix (BIO-RAD): *DLL4* (Hs00184092_m1), *HEY1* (Hs00232618_m1), *HEY2* (Hs00232622_m1), *HES1* (Hs00172878_m1), *EFNB2* (Hs00187950_m1), *NRARP* (Hs01104102_s1). VIC-conjugated *ACTB* (4352935) or VIC-conjugated *GAPDH* (44326317E) were used to normalize gene expression. HUVECs with at least three independent stimulation experiments for each gene were analyzed to obtain the relative expression differences using the $2^{-\Delta\Delta Ct}$ method.

## Mice and in vivo experiments

C57BL/6J and *Tg(Hey1-EGFP)$^{ID40Gsat}$* males or C57BL/6J at the age of 8 weeks were used for hydrodynamic tail vein injection. Animals were anesthetized with 3.75% isoflurane and injected with 5 µg/g (plasmid/body weight) plasmid suspended in TransIT-EE hydrodynamic delivery solution (MIR5340, Mirus) according to the manufacturer's instruction. The appropriate amount of plasmid was suspended in an injection volume of 10% of the body weight and injected into each individual mouse via the tail vein in 5–7 s as previously reported (*Liu et al., 1999*).

For the ovariectomy osteoporosis model, 8-week-old C57BL/6J female mice were anesthetized by intraperitoneally injection of ketamine hydrochloride (Ketavet; 100 mg/kg body weight) and xylazine (Rompun; 16 mg/kg body weight) in sterile PBS. Once animals had entered the tolerance phase without foot reflexes, a small incision was made along the dorsal midline, the abdominopelvic cavity was opened, and ovaries were removed bilaterally. The skin was closed by metal clips. After that, analgesic Carprofen (Rimadyl; 4 mg/kg body weight) was applied subcutaneously. Mice in the sham group underwent the same procedure without removal of ovaries. Four weeks after the surgery, mice received *pLIVE-Dll4(E12)* or vehicle by hydrodynamic tail vein injection. Three weeks later, mice were analyzed and the successful ovariectomy was confirmed by the reduction of uterus size.

For PTH and *pLIVE-Dll4$_{(E12)}$* administration, 8-week-old C57BL/6J females were divided randomly into four groups: vehicle control, *pLIVE-Dll4(E12)*, PTH, and PTH plus *pLIVE-Dll4(E12)*. Following *pLIVE-Dll4(E12)* injection at day 0, PTH (1-34) (human) acetate salt (Bachem) was injected subcutaneously

every day from day 1 to day 20 at a dose 100 mg/kg body weight, diluted in sterilized PBS. Mice were analyzed at day 21.

All animals were housed at the Max Planck Institute for Molecular Biomedicine and protocols were approved by animal ethics committees with permissions (Az 81–02.04.2019 .A114 and Az 81–02.04.2020 .A416) granted by the Landesamt für Natur, Umwelt und Verbraucherschutz (LANUV) of North Rhine-Westphalia.

## Bone sample preparation and immunohistochemistry

Freshly isolated femurs were collected from plasmid and vehicle-injected mice and immediately fixed in 4% ice-cold paraformaldehyde (PFA) solution for 24 hr. Decalcification was carried out with 0.5 M EDTA (pH 8.0) at 4°C with constant shaking on a horizontal shaker for 48 hr. Next, decalcified bones were immersed into 20% sucrose and 2% polyvinylpyrrolidone (PVP) for 24 hr. Finally, tissues were embedded and frozen in embedding solution containing 8% porcine gelatin together with 20% sucrose and 2% PVP. Cryosections of 80–100 µm thickness were generated with low-profile disposable blades 819 (LEICA) on a Leica CM2050 cryostat.

For immunostaining, bone sections were air-dried, rehydrated in PBS, permeabilized for 30 min with 0.3% Triton X-100 PBS and blocked in PBS containing 5% donkey serum at room temperature for 30 min. Blocked sections were incubated with primary antibodies diluted in 5% donkey serum in PBS overnight at 4°C. After primary antibody incubation, sections were washed with PBS for three times and incubated with appropriate Alexa-Fluor-conjugated secondary antibodies (1:400) diluted in PBS for overnight at 4°C. Nuclei were counterstained with DAPI after secondary antibody incubation. Sections were washed four times with PBS before mounting with FluoroMount-G (0100–01, Southern Biotech). Finally, the slides were air-dried and sealed with nail polish.

The following primary antibodies were used: rat anti-Endomucin (SC-65495, Santa Cruz, 1:100), rat anti-PECAM-1 (553370, Pharmingen), rabbit anti-Osterix (Abcam, ab22552, 1:1000), rat anti-Runx2 (MAB2006, R&D, 1:50), rabbit anti-Collagen type I (AB675P, Millipore, 1:200), goat anti-Osteopontin (AF808, R&D Systems, 1:100), rabbit anti-Osteocalcin (LS-C17044, LifeSpan BioSciences, 1:100), goat anti-Dll4 (AF1389, R&D Systems, 1:100), goat anti-Jag1 (J4127, Sigma, 1:100), goat anti-PDGFRβ (AF1042, R&D, 1:100), rabbit anti-NG2 (AB5320, Millipore), goat anti-BCAM (AF8299, R&D Systems, 1:50), rabbit anti-ATP6V1B1+ ATP6V1B2 (ab200839, Abcam), rabbit anti-Aggrecan (AB1031, Millipore), goat anti-Sox9 (AF3075, R&D Systems), rabbit anti-Perilipin (9349, Cell Signaling).

Secondary antibodies were: donkey anti-rat IgG conjugated to AF594 (A21209, Invitrogen), donkey anti-goat IgG conjugated to AF647 (A21447, Invitrogen), donkey anti-rabbit IgG conjugated to AF488 (A21206, Invitrogen), donkey anti-rabbit IgG conjugated to 594 (A21207, Invitrogen), donkey anti-rat IgG conjugated to 647 (712-605-153, Jackson ImmunoResearch), donkey anti-goat IgG conjugated to AF488 (A11055, Invitrogen), donkey anti-rabbit IgG conjugated to AF647 (A31573, Invitrogen), donkey anti-rat IgG conjugated to AF488 (A21208, Invitrogen), and donkey anti-goat IgG conjugated to AF647 (A32849, Thermo Scientific).

## Histology of liver and intestine

The dissected intestinal tract was flushed gently with cold PBS followed by a flush with 4% PFA. The intestine was fixed at 4°C for 16 hr. The median lobe of liver was dissected and fixed at 4°C for 16 hr. Formalin-fixed and paraffin-embedded tissues were sectioned at 7 µm thickness. Histochemical identification of intestinal cell types was performed with Alcian blue and Nuclear Fast Red double staining. Liver sections were stained with hematoxylin and eosin. Images were taken with Zeiss Axio Imager M1 microscope.

## Immunohistochemistry of lung

Lung sample preparation and immunohistochemistry were performed as previously described (*Kato et al., 2018*). To expose the heart and lungs, the chest cavity of anesthetized mice was opened. Through the right ventricle with manual pressure, a warm (37°C) solution of 6% gelatin/PBS (G1890, Sigma) was gently perfused. After 15 min exposure to ice-cold paper tissue, the ventral trachea was cannulated with an intravenous catheter tube that was secured by tying a suture around the trachea. The lungs were inflated to full capacity by gently injecting warm (37°C) 1% low gelling agarose/PBS (A4018, Sigma). After 20 min exposure to ice-cold paper tissue, the lungs were placed in 2% PFA

solution (PFA/PBS, 4°C, P6148, Sigma) for 30 min. After washing with cold PBS for 30 min, the lung lobes were sliced (150 µm) using vibrating blade microtome (VT1200, Leica). Lung slices were fixed in 4%PFA/PBS at 4°C for 1 hr, washed thoroughly in PBS and incubated twice in PBS for 30 min at room temperature. Lung slices were blocked (5% donkey serum, 0.5% Triton X-100 in PBS) overnight at 4°C. Next, sections were treated with primary antibodies (goat anti-CD31 (FAB3628, R&D Systems, 1:100), rat anti-CD31 (553370, Pharmingen, 1:100), goat anti-DLL4 (R&D Systems; AF1389, 1:100)) in blocking solution over night at 4°C. Following four washes with PBST (10 min each), sections were incubated with secondary antibodies (donkey anti-goat IgG AF488 [A11055, Invitrogen, 1:500], donkey anti-goat IgG AF647 [A21447, ThermoFisher, 1:500], donkey anti-rat IgG AF647 [A21247, Invitrogen, 1:500]) in blocking solution over night at 4°C. Nuclei were counterstained with DAPI (D9542, Sigma, 2 µg/mL). After four wash steps with PBS, sections were mounted using FluoroMount-G (Southern Biotech) under cover slips.

## Analysis of inflammatory cytokines and thymic T cells

Peripheral blood was collected into anticoagulant EDTA-treated Microvettes (20.1341, Sarstedt) for blood collection. Cells were removed from plasma by centrifugation for 15 min at 2000× $g$ using an ice-cold refrigerated centrifuge. Cytokines levels were measured from the resulting supernatant with LEGENDplex Mouse Inflammation Panel (13-plex) with V-bottom plates (740446, Biolegend). Analysis and quantification were performed according to the manufacturer's protocol. Data analysis was performed using software provided by Biolegend. Manual gating was used to define beads A and B, while an automatic gating strategy was used to gate individual cytokine in APC-PE plot.

Mice were euthanized for analysis at the age of 11 weeks. To prepare single cell suspensions of thymocytes, individual thymi were placed in pre-wet 70 µm cell strainers immersed in ice-cold 2% FCS/PBS in petri dishes for gentle disruption with the end of a 5 mL syringe plunger. Thymocytes were stained with rat-anti-CD45-FITC (11–0451, eBioscience), rat-anti-CD8-Biotin (13–0081, eBioscience), and rat-anti-CD4-APC (17-0042-82, eBioscience) followed by staining with Streptavidin PE/Cy7 (SA1012, Thermo Scientific). Flow cytometry was performed using a FACSAria Fusion with FACSDiva (BD Biosciences).

## Analysis of vascular leakage

Adult mice were injected with 1 mg of 70 kDa Dextran, Texas Red (Lysine fixable, D1864, Molecular Probes) dissolved in 200 µL of PBS via tail vein injection. Femur, liver, kidney, and lung were harvested from Dextran and PBS-injected mice after 15 min and subjected to sample preparation and staining as describe above for bone samples.

## Micro-CT analysis and histomorphometry

Tibiae were collected and attached soft tissue was removed thoroughly prior to fixation in 4% PFA overnight at 4°C. Fixed samples were analyzed with µ-CT 50 by Scanco Medical AG, Switzerland. A voxel size of 6 µm was chosen in all three spatial dimensions. For each sample, 500 slices were evaluated covering a total of 3 mm, X-ray voltage was 70 kVp, intensity 86 µA, and integration time 1000 ms, frame averaging 1.

Calcein double labeling was performed as reported previously (*Porter et al., 2017*) with minor modifications to calculate bone formation rate (BFR) and mineral apposition rate (MAR). Briefly, mice were intraperitoneally injected with 10 mg/kg calcein (Sigma, C0875) dissolved in 2% sodium bicarbonate solution at the tenth day and third day before euthanasia. Bones were fixed in 4% PFA for 2 days at room temperature followed by transfer to 70% ethanol. Fixed bones were incubated with 5% potassium hydroxide for 96 hr at room temperature. Processed bones were embedded in bone embedding solution (8% porcine gelatin in the presence of 20% sucrose and 2% PVP). Cryosections were taken at 20 µm using low-profile blades on a Leica CM2050 cryostat. Single plane images were acquired from the sections using an LSM 880 confocal microscope (Carl Zeiss). Representative images show trabecular bone. MAR and BFR were calculated from trabecular bone in the metaphysis.

## Single cell sequencing of BM stromal cells

Femurs and tibiae were harvested 3 wpi from adult mice at the age of 11 weeks. Bones were crushed by pestle and mortar with ice-cold 2% FCS/PBS and followed by 30 min digestion at 37°C with

collagenase cocktail by mixing with 2 mg/mL of collagenase IV (17104–019, Gibco), 2 mg/mL collagenase I (17100–017, Gibco), and 10 U/mL of Dispase (17105–041) in a 1:1:1 ratio. The resulting single cell suspension was subjected to depletion of lineage-positive hematopoietic cells with following reagents from Miltenyi Biotec: Lineage Cell Depletion Kit (130-090-858), CD45 Microbeads (130-052-301), CD117 Microbeads (130-091-224), Ter-119 Microbeads (130-049-901), and CD71-biotin (Biolegend, 113803) in combination with Anti-Biotin UltraPure MicroBeads (Miltenyi Biotec, 130-105-637). Lineage-depleted stromal cells were loaded into a BD Rhapsody Cartridge for capturing single cells for whole transcriptome according to the manufacturer's instructions. scRNA libraries were prepared with The BD Rhapsody Whole Transcriptome Analysis (WTA) Amplification kit. DNA sequencing was performed on a NextSeq500 (Illumina).

## scRNA-seq bioinformatics and data access

FASTQ files from BD Rhapsody WTA and sequencing were first trimmed for adapters and Phred score (>20) using TrimGalore! (0.6.4), discarding reads shorter than 66 bp after trimming. The first mate was then split into barcode and UMI using the pattern described in the BD Rhapsody Bioinformatics Handbook (revision 6.0, *Figure 3*), allowing up to two mismatches in each of the fixed sequences L1 and L2. A barcode whitelist was created by counting the occurrences of each of the three CLS sequences, choosing those that occur at least 1000 times, and forming every possible combination of CLS1-CLS2-CLS3. Reads were mapped using Salmon alevin (1.1.0) with the vM22 Gencode transcripts and genomic decoy sequences, including the above whitelist for barcode correction. Automatic filtering by Salmon alevin was disabled, and Alevin count matrices were used for downstream analyses. Count matrices were loaded into R using tximport (1.1.14). Empty wells were identified using DropletUtils (1.6.1) with an FDR cutoff of 0.001. Cells classified as empty wells or with less than 500 expressed genes, and genes with signal in less than 10 cells were removed from the analysis. Doublets were identified using scDblFinder (1.0.0) with default parameters and subsequently removed. Filtered count matrices were imported into Seurat (3.1.3). Mitochondrial gene contribution was estimated using Seurat's PercentageFeatureSet, and cells with more than 20% mitochondrial reads were removed. Expression counts were normalized using scran normalize (1.14.6). Control and treatment samples were analyzed together as an integrated Seurat dataset. The data was scaled using Seurat's ScaleData function and variable features were selected by FindVariableFeatures (nFeatures = 2000, selection.method='vst').

Main cell type clusters were identified using Louvain clustering with a resolution parameter of 0.05, using the top 50 principal components (PCs). Marker genes were identified using Seurat's FindAllMarkers function with default settings, and cells identified as hematopoietic were removed as contaminants. Marker expression was plotted using DoHeatmap on the top 10 marker genes in each cluster. Dimensionality reduction plots are based on UMAP projections using Seurat's RunUMAP function.

For the MSC population subclustering, we first applied stricter empty droplet filtering (FDR = 0) to remove low complexity subpopulations. The expression values were re-scaled and variable feature selection was repeated. We then used the top six PCs for Louvain clustering at a resolution of 0.05 and UMAP projection. Marker genes were identified using Seurat's FindAllMarkers function with default settings, and marker expression was plotted using DoHeatmap on the top five marker genes in each cluster.

Differentially expressed genes were identified separately for each cluster, using Seurat's FindMarkers function with default settings (Wilcoxon rank sum test) on the $Dll4_{(E12)}$-treated and control samples. Scaled expression violin plots are based on Seurat's RNA assay and the 'data' slot. Each violin is scaled to the width of the plotting window.

GO analysis was performed using GOstats (2.52.0) hypergeometric testing. p-Values were multiple testing corrected using the Bonferroni method.

scRNA-seq data have been deposited in the GEO functional genomics data repository under the accession number GSE152285.

## Western blots

Half of the median liver lobe was dissected from plasmid and vehicle injected mice and snap-frozen in liquid nitrogen. Liver samples were homogenized in lysis buffer (50 mM Tris-HCl pH 7.4, 1 mM EDTA, 1% Triton X-100, 0.2% Na-DOC, 0.2% SDS, cOmplete ULTRA Proteinase inhibitor 1× [Roche,

05892970001], Phosphatase Inhibitor Cocktail Set V [524632-1SET, Merck], 1 mM PMSF) with Tissue-lyser LT (QIAGEN) and clarified by centrifugation by 20,000× $g$ for 20 min at 4°C. Protein concentration in lysates was measured using Pierce BCA Protein-Assay kit (23225, Pierce). Soluble supernatants were prepared in SDS-PAGE sample buffer and analyzed by SDS-PAGE and immunoblotting after loading 40 µg of total liver lysate. Signal was detected using horseradish peroxidase-conjugated secondary antibodies followed by ECL Prime Western Blotting Detection Reagents (RPN2232, GE Healthcare). Primary antibodies: goat anti-Dll4 (AF1389, R&D, 1:200), rabbit anti-GAPDH (AM4300, Ambion, 1:1000). Secondary antibodies: goat anti-rabbit IgG, HRP-linked whole Ab (7074, Cell Signaling, 1:15000), donkey anti-goat IgG, HRP-linked whole Ab (ABIN1536502, antibody online, 1:15,000).

0.5 µg recombinant Jag1 recombinant proteins were prepared in SDS-PAGE sample buffer and analyzed by SDS-PAGE, followed by immunoblotting. Primary antibodies used are mouse anti-His tag (Zymed; 372900 1:200) and goat anti-Jag1 (Sigma; J4127 1:500). Secondary antibodies used are sheep anti-mouse IgG 656G, HRP-linked whole Ab (HG-Healthcare, NA931, 1:40000) and Peroxidase AffiniPure bovine anti-goat IgG (H + L) (Jackson ImmunoResearch, 805-035-180, 1:15,000).

### Image acquisition, processing, and statistical analysis

Confocal image acquisition was performed using confocal microscope LSM880 (Zeiss). Z-stacks of images were processed and 3D reconstructed with Imaris software (version 9.50, Bitplane). Image J (NIH) and Illustrator (Adobe) software were used for image processing. Quantifications of vascular and bone-related parameters were performed with Fiji software on high-resolution images.

For quantification of OSX$^+$ and Runx2$^+$ cells, a region of 400 µm from growth plate toward the caudal region was selected in images of the metaphysis. Osteoclast surface/bone surface (Oc. S/BS; %) and osteoclast number/bone perimeter (No. Oc./B. Pm) were calculated based on ATP6V1B1 + ATP6V1B2 staining of bone sections. VEGFR3$^{low}$ area was calculated based on VEGFR3 staining and normalized to the total selected area. Adipocyte cell number was calculated based on Perilipin staining of bone sections. Artery number was calculated based on BCAM staining.

Col1a1, PDGFRb, Dll4, Endomucin, Opn areas were calculated based on the corresponding stainings and were normalized to DAPI area. The length of metaphysis was calculated as the mean of three values: length of left, right, and middle regions of metaphysis.

All images shown are representative for the respective staining in several experiments. Within one experiment laser excitation and confocal scanner detection were the same. All images shown in the figures are maximum intensity projections and are representative of at least three mice analyzed for each condition unless stated otherwise.

Statistical analysis was performed with GraphPad Prism software. All data are presented as mean ± s.e.m. or mean ± SD. Unpaired two-tailed Student's t-test, Mann-Whitney U test, one-way and two-way ANOVA were used to determine statistical significance, as indicated in the legends. All experiments were performed independently at least three times and respective data were used for statistical analysis. Sample sizes for each experiment are described in the respective figure legends. No randomization or blinding was used and no animals were excluded from analysis. Several independent experiments were performed to guarantee reproducibility of findings.

## Acknowledgements

The authors thank the Max Planck Society, the European Research Council (AdG 339409 AngioBone; AdG 786672 PROVEC), and the Leducq Foundation for funding. ECW was supported by the Alexander von Humboldt Foundation.

## Additional information

### Funding

| Funder | Grant reference number | Author |
| --- | --- | --- |
| Max Planck Society | | Ralf H Adams |

| Funder | Grant reference number | Author |
|---|---|---|
| European Research Council | AdG 339409 AngioBone | Ralf H Adams |
| European Research Council | AdG 786672 PROVEC | Ralf H Adams |
| Leducq Foundation | | Ralf H Adams |

The funders had no role in study design, data collection and interpretation, or the decision to submit the work for publication.

## Author contributions

Cong Xu, Conceptualization, Investigation, Methodology, Writing – original draft; Van Vuong Dinh, Emma C Watson, Susanne Adams, Frank Berkenfeld, Martin Stehling, Seyed Javad Rasouli, Rui Fan, Rui Chen, Ivan Bedzhov, Qi Chen, Katsuhiro Kato, Investigation, Methodology; Kai Kruse, Investigation, Project administration, Methodology; Hyun-Woo Jeong, Investigation, Project administration, Methodology; Mara E Pitulescu, Conceptualization, Investigation, Project administration, Supervision, Writing – review and editing; Ralf H Adams, Conceptualization, Supervision, Investigation, Project administration, Supervision, Writing – original draft, Writing – review and editing

## Author ORCIDs

Emma C Watson (ID) http://orcid.org/0000-0002-0986-5524
Qi Chen (ID) http://orcid.org/0000-0001-8485-6540
Mara E Pitulescu (ID) http://orcid.org/0000-0001-5322-8146
Ralf H Adams (ID) http://orcid.org/0000-0003-3031-7677

## Ethics

All animals were housed at the Max Planck Institute for Molecular Biomedicine and protocols were approved by animal ethics committees with permissions (Az 81-02.04.2019.A114 and Az 81-02.04.2020. A416) granted by the Landesamt für Natur, Umwelt und Verbraucherschutz (LANUV) of North Rhine-Westphalia. Every effort was made to minimize suffering.

## Decision letter and Author response

Decision letter https://doi.org/10.7554/eLife.60183.sa1
Author response https://doi.org/10.7554/eLife.60183.sa2

# Additional files

## Supplementary files
• Transparent reporting form

## Data availability

scRNA-seq data have been deposited in the GEO functional genomics data repository under the accession number GSE152285.

The following dataset was generated:

| Author(s) | Year | Dataset title | Dataset URL | Database and Identifier |
|---|---|---|---|---|
| Adams RH, Xu C, Kruse K, Jeong H, Watson E, Adams S, Stehling M, Chen Q, Kato K | 2020 | Induction of osteogenesis by bone-targeted Notch activation | https://www.ncbi.nlm.nih.gov/geo/query/acc.cgi?acc=GSE152285 | NCBI Gene Expression Omnibus, GSE152285 |

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
