## [Editor Report]

Osteoporosis most often treated by reducing bone resorption as there are limited choices of medication that are anabolic for bone. Previous studies have suggested that the Notch signaling pathway could be targeted to enhance osteogenesis. The authors have intriguingly generated a soluble bone targeted fusion protein comprised of a modified version of the extra-cellular domain of the Delta-like 4 Notch ligand and poly-aspartate peptide motif with binding affinity for hydroxyapatite in the bone matrix. This approach has high potential as a future therapeutic for osteoporosis.

---

## [Decision Letter]

**Decision letter after peer review:**

Thank you for submitting your article "Induction of osteogenesis by bone-targeted Notch activation" for consideration by *eLife*. Your article has been reviewed by 3 peer reviewers, and the evaluation has been overseen by a Reviewing Editor and Clifford Rosen as the Senior Editor. The reviewers have opted to remain anonymous.

The reviewers have discussed the reviews with one another and the Reviewing Editor has drafted this decision to help you prepare a revised submission.

Summary:

Osteoporosis most often treated by reducing bone resorption as there are limited choices of medication that are anabolic for bone. Previous studies have suggested that the Notch signaling pathway could be targeted to enhance osteogenesis. In this study, the author have generated a soluble bone targeted fusion protein comprised of a modified version of the extra-cellular domain of the Delta-like 4 Notch ligand and poly-aspartate peptide motif with bind affinity for hydroxyapatite. Using a combination of single sequencing and histology, the authors show that this fusion protein can induce bone formation in mice and limited mechanistic information was learned. That said, this approach has high potential as a future therapeutic for osteoporosis. While the approach was considered novel, all reviewers agreed that this paper required additional data to demonstrate the true therapeutic potential of this fusion protein as a bone anabolic. In several instances requests for clarification were made. These comments are summarized below.

Essential Revisions:

1. All reviewers requested a relevant therapeutic model be employed to demonstrate that this fusion protein would work in the face of low bone mass. While several models (tail suspension, aging, etc) were proposed, it was agreed upon that an ovariectomy model might be the easiest and most relevant model in which to demonstrate effectiveness after bone loss. This was considered to be an essential experiment by all three reviewers. Even if this fusion protein fails to rescue bone after ovariectomy, this would be very import knowledge for the field and would increase our understanding about Notch signaling in bone.

2. Please show immunoreactivity of the liver expressed constructs in other organs than bone. This should include at minimum, the liver and spleen.

3. The dextran texas red showed that there is no barrier leakage in bone. This should also be shown in other highly vascularized organs.

4. The authors show immunoreactivity in the bone marrow of the Jag1-Asp8 in Figure 5A, but less for the Dll4-Asp8 construct. This point needs clarification and or additional explanation.

5. Since there is significant targeting of the cortical bone one would expect an effect in osteocytes. Surprisingly there was limited evidence of Notch activation in RNASeq studies. Could the lack of osteocyte signatures could be due to the method used for cell preparation? This requires clarification in methods, limitations and in the manuscript discussion.

6. In Figure 5, the authors claim that JAG1 activating ligand has no impact on bone mass; however, they do not show whether Notch is truly activated by this larger size ligand in vitro or in vivo. This was performed for DLL4 and could be similarly done for JAG1, as it may shed light on whether insufficient Notch signaling is observed within bone associated cells.

7. Due to the fact that the authors only assess phenotypes 3 weeks post-injection, it is difficult to separate direct vs. indirect effects or cell autonomous vs. non-autonomous effects of the cell specific Notch activation. One assessment that would help elucidate this would be to determine the overlap (Figure 7) of Hey1 and Sp7 double positive cells to determine whether active Notch signaling is promoting osteoblastic differentiation. Alternatively, the majority of Sp7+ cells may be Hey1 negative and therefore lead to the conclusion that Notch signaling must be down-regulated in order to achieve osteoblast differentiation, and may in fact be maintaining and expanding MSC populations that later go on to form osteoblasts. Alternatively or additionally, a more immediate time point could be assessed from Hey1-GFP mice injected with DLL4. GFP+ and GFP- MSCs could be FACS isolated and compared via qPCR for osteogenic and MSC markers. This may also give a better indication of how DLL4-mediated Notch activation is affecting MSCs in a cell autonomous manner.

8. The authors describe the effects of a modified Dll4-Asp8x fusion protein targeted to bone and demonstrate an increase in bone volume/total volume (BV/TV) due to an increase in trabecular thickness and bone formation. This is somewhat unexpected because 1. Asp8 usually targets bone resorbing surfaces whereas AspSerSerx6 targets bone forming surfaces (Zhang Nat Med 18;307,2012), 2. Activation of Notch signaling has been shown to inhibit osteogenesis and 3. Notch signaling affects osteoclastogenesis, albeit this is dependent on the Notch receptor activated and Dll4 would activate all Notch receptors. These points need clarification in this manuscript.

9. The authors do not report whether the animals studied were littermate and sex matched, This is a critical point as the bone microarchitecture of male and female mice is substantially different.

10. The data presented for the bone histomorphometry and uCT are incomplete. The authors show an effect on bone formation but do not provide data on osteoblast surface or number. Similarly, there is no effect on osteoclast number, but eroded surface is not shown. There is significant presence of Dll4 in cortical bone but uCT data on cortical bone are not provided. These data are essential for transparent interpretation of this study results.

11. Increased expression of markers of osteoblast differentiation and function is somewhat limited to Osx. Data on osteocalcin and proteoglycan were not quantified and immunohistochemistry does not reveal a pronounced effect. Please clarify.

12. For the RNA Seq data, it is surprising that bone marrow stromal (BMS) cells are affected whereas osteoblasts are not since BMS differentiate into osteoblasts and alterations in BMS cells would not explain an increase in mineral apposition rate unless they differentiate into osteoblasts. The markers of osteogenesis affected are not the classically defined osteogenic factors (IGF2, BP3, Prg4) and one needs to be cautious with the interpretation of these results and with the attribution of the effect to BMS cells.

13. The authors suggest possible therapeutic value of Dll4 fusion proteins but what was administered to the animals of this study was a construct expressing the protein in vivo. A more practical and convincing approach would be the injection of the protein itself. This should at minimum be discussed as a future-directions.

---

## [Author Response]

Essential revisions:1. All reviewers requested a relevant therapeutic model be employed to demonstrate that this fusion protein would work in the face of low bone mass. While several models (tail suspension, aging, etc) were proposed, it was agreed upon that an ovariectomy model might be the easiest and most relevant model in which to demonstrate effectiveness after bone loss. This was considered to be an essential experiment by all three reviewers. Even if this fusion protein fails to rescue bone after ovariectomy, this would be very import knowledge for the field and would increase our understanding about Notch signaling in bone.

As suggested by the reviewers, we have now included in revised Figure 6 a detailed characterization of sham operated versus ovariectomized (OVX) female mice, injected with PBS or *pLIVE-Dll4_(E12)_* plasmid. We have performed micro-CT (revised Figure 6A and B) and provide quantitation of multiple parameters such as BV/TV, connectivity density, trabecular thickness, cortical consistency, trabecular separation, bone surface, trabecular number, and cortical thickness in revised Figure 6C and D.

Although the ovariectomy was successful and resulted in bone loss (revised Figure 6 A-D, G), *pLIVE-Dll4_(E12)_* injection did not increase bone deposition in OVX females, as indicated by the micro-CT and histomorphometry analysis (revised Figure 6 A-D) as well as Col1a1 and PDGFRβ immunostaining (revised Figure 6E and F). Furthermore, we found that Dll4_(E12)_ has no significant effect on bone formation in sham operated females (revised Figure 6A-D). Considering that our previous experiments with male mice resulted in significantly increased bone formation (revised Figure 1), the absence of beneficial effects in female animals is most likely caused by lower bone surface and therefore decreased Dll4_(E12)_ binding relative to males (revised Figure 6G and Figure 6 suppl. 1). This could have been a somewhat disappointing result, but, fortunately, we found in further experiments that Dll4_(E12)_ can have beneficial effects even in female mice. PTH administration to osteoporotic patients is used to increase bone formation (Augustine and Horwitz, Current Osteoporosis Report 2013, PMID: 24078470) and similar effects are seen in rodents (Li et al., Bone 1999, PMID: 9951776; Wein et al., CSH Perspective in Medicine 2018, PMID: 29358318). Therefore, we decided to combine Dll4_(E12)_ and PTH administration to female mice and compare the results to single treatment groups and untreated controls, which revealed striking additive and, for some parameters, synergistic effects of Dll4_(E12)_ and PTH on bone formation. These exciting new results are presented in revised Figure 7 (micro-CT data), revised Figure 7 suppl. 1 (Dll4 immunoreactivity, and Col1a, PDGFRβ and Emcn immunostaining), and in revised Figure 7 suppl. 2 (Osx and Opn immunostaining).

2. Please show immunoreactivity of the liver expressed constructs in other organs than bone. This should include at minimum, the liver and spleen.

Thank you for this suggestion. In revised Figure 1 suppl. 5, we now present Dll4 immunostaining for liver, spleen and lung, which indicates that there is no accumulation of Dll4_(E12)_ in these organs. This confirms our overall experimental strategy and explains why effects of Dll4_(E12)_ are confined to the skeletal system.

3. The dextran texas red showed that there is no barrier leakage in bone. This should also be shown in other highly vascularized organs.

We followed the reviewers’ suggestion and provide in revised Figure 2—figure supplement 2 A-C images and quantitation of Dextran Texas Red signal per tissue area in liver, spleen and lung of control versus Dll4_(E12)_-injected mice. These results show that there is no significant difference in Dextran leakage relative to control in all organs investigated.

4. The authors show immunoreactivity in the bone marrow of the Jag1-Asp8 in Figure 5A, but less for the Dll4-Asp8 construct. This point needs clarification and or additional explanation.

Thank you very much for this question. First of all, we need to emphasize that a direct comparison of these immunostainings is difficult because two different antibodies (anti-Dll4 and anti-Jag1, respectively) have been used. As they are binding different epitopes, these reagents are likely to have different properties with implications for staining intensity, specificity and background.

If one assumes that stainings predominantly reflect the distribution pattern of the Asp_8x_-coupled Notch ligands, Dll4-Asp_8x_ (Figure 1—figure supplement 3A), Dll4_(E12)_ (Figure 1B), and, to some extent, also Jag1_(JV1)_ (Figure 5—figure supplement 2A) indeed appear to be more strongly confined to bone surface than Jag1-Asp_8x_ (Figure 5—figure supplement 2A). These differences might reflect distinct modes of binding to different Notch receptors, which are known to be different for Jagged1 relative to Dll4 and also depend on receptor glycosylation (e.g. Luca et al., Science 2017, PMID: 28254785). However, it has to be noted that Jag1-Asp_8x_ staining can be also seen at the surface of bone, indicating that the protein is present at the right place but does not show the same biological activity as Dll4_(E12)_. We can only speculate about the underlying reasons, but, presumably, only Dll4_(E12)_ achieves the threshold of Notch activation leading to an increase in bone.

We now mention this aspect in the Discussion but have avoided overly speculative statements. We also clarify that our findings do not address the function of endogenous Jagged1, which has been previously linked to the regulation of osteogenic differentiation.

5. Since there is significant targeting of the cortical bone one would expect an effect in osteocytes. Surprisingly there was limited evidence of Notch activation in RNASeq studies. Could the lack of osteocyte signatures could be due to the method used for cell preparation? This requires clarification in methods, limitations and in the manuscript discussion.

As the reviewer points out correctly, there is indeed a lack of osteocytes in our scRNA-seq data, likely due to the fact that our collagenase-based digestion method could not release these cells from their lacunae in bone. This point is now mentioned in the revised manuscript.

6. In Figure 5, the authors claim that JAG1 activating ligand has no impact on bone mass; however, they do not show whether Notch is truly activated by this larger size ligand in vitro or in vivo. This was performed for DLL4 and could be similarly done for JAG1, as it may shed light on whether insufficient Notch signaling is observed within bone associated cells.

As suggested by the reviewers, we present in revised Figure 5—figure supplement 2 new in vitro data showing the purification of Jag1-Asp_8x_ and Jag1_(JV1)_ proteins as well as the immunoblot detection of those protein by anti-Jag1 antibody. Both immobilized or in solution, Jag1-Asp_8x_ and Jag1_(JV1)_ proteins are able to affect Notch target gene expression in cultured cells (revised Figure 5—figure supplement 1D, E). The changes in transcript expression triggered by the two recombinant Jag1 proteins follow a similar trend but, as expected, the Jag1_(JV1)_ variant is more potent in these assays.

7. Due to the fact that the authors only assess phenotypes 3 weeks post-injection, it is difficult to separate direct vs. indirect effects or cell autonomous vs. non-autonomous effects of the cell specific Notch activation. One assessment that would help elucidate this would be to determine the overlap (Figure 7) of Hey1 and Sp7 double positive cells to determine whether active Notch signaling is promoting osteoblastic differentiation. Alternatively, the majority of Sp7+ cells may be Hey1 negative and therefore lead to the conclusion that Notch signaling must be down-regulated in order to achieve osteoblast differentiation, and may in fact be maintaining and expanding MSC populations that later go on to form osteoblasts. Alternatively or additionally, a more immediate time point could be assessed from Hey1-GFP mice injected with DLL4. GFP+ and GFP- MSCs could be FACS isolated and compared via qPCR for osteogenic and MSC markers. This may also give a better indication of how DLL4-mediated Notch activation is affecting MSCs in a cell autonomous manner.

We are grateful for these comments and agree that it is important to understand how Dll4_(E12)_ promotes osteogenesis. Various lines of evidence indicate that the treatment primarily affects immature (OSX-negative) mesenchymal stromal progenitor cells:

1) Dll4_(E12)_ leads to increased expression of *Hey1-GFP* signal in vessel-associated in the metaphysis and adjacent transition zone to the diaphysis. These cells show only weak or no nuclear expression of Osterix (Figure 3—figure supplement 1).

2) Our scRNA-seq data shows profound Dll4_(E12)_-induced gene expression changes in diaphyseal mesenchymal stromal cells (dpMSCs) but not in osteoblasts. This argues that, while we see an increase in OSX+ cells, gene expression in this population is not substantially altered by Dll4_(E12)_ (revised Figure 8F-G and Figure 8—figure supplement 3D, E).

3) Previous work by us and others has shown that vessel-associated mesenchymal stromal cells in the diaphysis can give rise to bone-forming cells (Zhou et al., Cell Stem Cell 2014, PMID: 24953181; Mizoguchi et al., Dev Cell 2014, PMID: 24823377; Sivaraj et al., Cell Rep 2021, PMID: 34260921).

Altogether, our results indicate that Dll4_(E12)_ activates Notch signaling primarily in immature mesenchymal stromal cells (revised Figure 9G), leads to the expansion of this population (Figure 3—figure supplement 2A, B) and, consistent with earlier studies (Hilton et al., Nat Med 2008, PMID: 18297083), reduced Runx2 expression (revised Figure 3B). This is fully consistent with previous findings showing that genetic approaches leading to Notch activation in mesenchymal progenitors suppress differentiation and promote the expansion of these cells (Hilton et al., Nat Med 2008, PMID: 18297083; Engin et al., Nat Med 2008, PMID: 18297084; Canalis et al., Endocrinology 2013, PMID: 23275471). In this context, it is presumably highly advantageous that Dll4_(E12)_-mediated Notch activation is, in contrast to genetic approaches, transient and does not suppress osteoblastic differentiation, presumably because cells are not permanently exposed to the bone surface-bound recombinant protein. This is also supported by the absence of *Hey1-GFP* signal in OSX+ cells (Figure 3—figure supplement 1) and lack of major transcriptional alterations in osteoblasts (Figure 8—figure supplement 2E). These findings might be conceptually relevant for the development of future strategies aiming at improved bone formation. We have added a paragraph covering the points above to the Discussion and trust that this addresses the question about the underlying mechanism raised by the reviewer.

8. The authors describe the effects of a modified Dll4-Asp8x fusion protein targeted to bone and demonstrate an increase in bone volume/total volume (BV/TV) due to an increase in trabecular thickness and bone formation. This is somewhat unexpected because 1. Asp8 usually targets bone resorbing surfaces whereas AspSerSerx6 targets bone forming surfaces (Zhang Nat Med 18;307,2012), 2. Activation of Notch signaling has been shown to inhibit osteogenesis and 3. Notch signaling affects osteoclastogenesis, albeit this is dependent on the Notch receptor activated and Dll4 would activate all Notch receptors. These points need clarification in this manuscript.

Thank you very much for these questions. There is indeed a substantial amount of literature describing different bone-homing moieties including the one mentioned in your comment. Other publications, such as Kusagi et al., (JBMR 2010, PMID: 10804024) and Miller et al., (Pharm. Res. 2008, PMID: 18758923), show that poly-Asp peptides bind different regions in long bone including the primary spongiosa in proximity of the growth plate. This property of poly-Asp peptides was appealing to us because we were initially trying to activate Notch signaling in the metaphyseal endothelium and, in fact, our own data also supports the deposition of Asp_8x_-fused proteins in the distal metaphysis (Figure 1—figure supplement 2, Figure 1—figure supplement 3A and Figure 5—figure supplement 2). We appreciate that other bone-homing peptides and reagents might provide different and perhaps even potent biological properties. This question will be an interesting topic for future work but extends beyond the scope of the current study, which establishes that Notch ligand fusion proteins containing bone-homing peptide sequences can be used to promote osteogenesis without adverse effects in other organs. We now mention the properties of different bone-homing epitopes in the Discussion.

Potential effects on osteoclastogenesis were not investigated further because vATPase+ osteoclasts were not significantly different between Dll4_(E12)_-treated and control mice.

9. The authors do not report whether the animals studied were littermate and sex matched, This is a critical point as the bone microarchitecture of male and female mice is substantially different.

We agree that this is a very important aspect. As discussed above under point 1, we had initially performed all experiments with male mice. In our revision, additional experiments revealed that females show less binding of recombinant Dll4, which is consistent with the lower bone surface in these mice relative to males (revised Figure 6 and Figure 6—figure supplement 1). Remarkably, the combination of Dll4_(E12)_ and PTH yields very strong effects in female animals, which indicates, for some of the parameters analyzed, synergy between the two treatments (revised Figure 7A, B). The sex of the animals in specific experiments is now indicated in the manuscript and figure legends.

10. The data presented for the bone histomorphometry and uCT are incomplete. The authors show an effect on bone formation but do not provide data on osteoblast surface or number. Similarly, there is no effect on osteoclast number, but eroded surface is not shown. There is significant presence of Dll4 in cortical bone but uCT data on cortical bone are not provided. These data are essential for transparent interpretation of this study results.

We are presenting in revised Figure 1E the data on osteoblast surface/ bone surface (Ob.S/BS (%)) as well as the graph on eroded surface/ bone surface (ES/BS (%)). Unfortunately, we don’t have micro-CT results for cortical bone in this data set. However, micro-CT analysis of female cortical bones show no benefit of Dll4_(E12)_ or the combination of Dll4_(E12)_ and PTH relative to vehicle control of PTH alone, respectively (revised Figure 7H-K).

11. Increased expression of markers of osteoblast differentiation and function is somewhat limited to Osx. Data on osteocalcin and proteoglycan were not quantified and immunohistochemistry does not reveal a pronounced effect. Please clarify.

We appreciate reviewers’ comment and have added quantitation graphs for Opn, Osteocalcin immunostained area in control and Dll4_(E12)_ treated mice in revised Figure 3. Additional quantitation for PDGFRβ, NG2, Acan and *Sox9* covered area are now shown in revised Figure 3—figure supplement 2.

12. For the RNA Seq data, it is surprising that bone marrow stromal (BMS) cells are affected whereas osteoblasts are not since BMS differentiate into osteoblasts and alterations in BMS cells would not explain an increase in mineral apposition rate unless they differentiate into osteoblasts. The markers of osteogenesis affected are not the classically defined osteogenic factors (IGF2, BP3, Prg4) and one needs to be cautious with the interpretation of these results and with the attribution of the effect to BMS cells.

In our view, the sum of our data indicates that Dll4_(E12)_ acts on immature progenitors, which increase in number, give rise to more osteoblasts and thereby enhance bone formation. Profound transcriptional changes are absent in osteoblasts, indicating that Dll4_(E12)_ increases the number but does not alter the molecular properties of these cells.

We agree that it is not clear to which extent alterations in various other pathways might contribute to the effect of Dll4_(E12)_ and we have put more emphasis on the role of Notch activation in mesenchymal stromal cells in the revised manuscript.

13. The authors suggest possible therapeutic value of Dll4 fusion proteins but what was administered to the animals of this study was a construct expressing the protein in vivo. A more practical and convincing approach would be the injection of the protein itself. This should at minimum be discussed as a future-directions.

We agree that the injection of recombinant proteins could be a next logical step in the development of potential new treatments. At the same time, the ongoing pandemic has recently shown that viral vectors or mRNA-based therapeutic approaches can play very important roles and such treatments are under development for various diseases apart from COVID19.

The aim of the current study was to address whether recombinant, bone-homing Notch ligands can be, in principle, used to increase bone formation. We have tested various constructs and might generate new versions in the future, which was greatly facilitated by the fact that we did not have to produce and purify large amounts of recombinant protein for each candidate. In this regard, our study also shows how candidate therapeutics can be developed and tested without excessive costs.